# STREAM-LEVEL FLOW MATCHING FROM A BAYESIAN DECISION THEORETIC PERSPECTIVE

## ABSTRACT

Flow matching (FM) is a family of training algorithms for fitting continuous normalizing flows (CNFs). Conditional flow matching (CFM) exploits the fact that the marginal vector field of a CNF can be learned by fitting least-square regression to the so-called conditional vector field specified given one or both ends of the flow path. We show that viewing CFM training from a Bayesian decision theoretic perspective on parameter estimation opens the door to generalizations of CFM algorithms. We propose one such extension by introducing a CFM algorithm based on defining conditional probability paths given what we refer to as "streams," instances of latent stochastic paths that connect pairs of noise and observed data. Further, we advocate the modeling of these latent streams using Gaussian processes (GPs). The unique distributional properties of GPs, and in particular the fact that the velocity of a GP is still a GP, allows drawing samples from the resulting stream-augmented conditional probability path without simulating the actual streams, and hence the "simulation-free" nature of CFM training is preserved. We show that this generalization of the CFM can substantially reduce the variance in the estimated marginal vector field at a moderate computational cost, thereby improving the quality of the generated samples under common metrics. Additionally, we show that adopting the GP on the streams allows for flexibly linking multiple related training data points (e.g., time series) and incorporating additional prior information. We empirically validate our claim through both simulations and applications to image and neural time series data.

## 1 INTRODUCTION

Deep generative models aim to estimate and sample from an unknown probability distribution. Continuous normalizing flows (CNFs, Chen et al. (2018)) construct an invertible and differentiable mapping, using neural ordinary differential equation (ODE), between a source and the target distribution. However, traditionally, it has been difficult to scale CNF training to large datasets (Chen et al., 2018; Grathwohl et al., 2019; Onken et al., 2021). Recently, Lipman et al. (2023); Albergo & Vanden-Eijnden (2023); Liu et al. (2023b) showed that CNFs can be trained via a regression objective, and proposed the flow matching (FM) algorithm. The FM exploits the fact that the marginal vector field inducing a desired CNF can be learned through a regression formulation, approximating per-sample conditional vector fields using a smoother such as a deep neural network (Lipman et al., 2023). In the original FM approach, the training objective is conditioned on samples from the target distribution, and the source distribution has to be Gaussian. This limitation was later relaxed, allowing the target distribution to be supported on manifolds (Chen & Lipman, 2024) and the source distribution to be non-Gaussian (Pooladian et al., 2023). Tong et al. (2024a) provided a unifying framework with arbitrary transport maps by conditioning on both ends. While their framework is, in principle, general, it does require the induced conditional probability paths be readily sampled from, and as such they considered a few Gaussian probability paths. Moreover, most existing FM methods only consider the inclusion of two endpoints, and hence cannot accommodate data involving multiple related observations, such as time series and other data with a grouping structure. Recently, Albergo et al. (2024) proposed a multimarginal stochastic interpolants, which can jointly learn a multivariate distribution by generalization of the stochastic interpolant framework. However, this approach may be too restrictive for path design, and hence may limit it usage to such as time series data.

In this paper, we first view FM from the perspective of Bayesian estimation under squared error loss, which motivates us to go one level deeper in Bayesian hierarchical modeling and specify distributional assumptions on *streams*, which are latent stochastic paths connecting the two endpoints. This leads to a class of CFM algorithms that conditions at the "stream" level, which broadens the range of conditional probability paths allowed in CFM training. By endowing the streams with Gaussian process (GP) distributions, these algorithms provide wider sampling coverage over the support of the marginal vector field, leading to reduced variance in the estimated vector field and improved synthetic samples from the target distribution. Furthermore, conditioning on GP streams allows for flexible integration of related observations through placing them along the streams between two endpoints and for incorporating additional prior information, all while maintaining analytical tractability and computational efficiency of CFM algorithms.

In summary, the main contributions of this paper are:

1. We present a Bayesian decision theoretic perspective on FM algorithms, which provides an additional justification for FM algorithms beyond gradient matching and serves as the foundation for extensions to these algorithms by latent variable modeling on the streams.

2. We generalize CFM training by augmenting the specification of conditional probability paths through latent variable modeling on the streams. We show that streams endowed with GP distributions lead to a simple stream-level CFM algorithm that preserves the "simulation-free" training.

3. We demonstrate that adjusting the GP streams can reduce the variance of the estimated marginal vector field with moderate computational cost.

4. We demonstrate how to use GP streams to integrate related observations, thereby taking advantage of the correlation among related samples to enhance the quality of the generated samples from the target distributions.

5. These benefits are illustrated by simulations and applications to image (CIFAR-10, MNIST and HWD+) and neural time series data (LFP), with code for Python implementation in the supplementary materials.

## 2 A Bayesian Decision Theoretic Perspective on Flow Matching

We start by viewing FM training from a Bayesian decision theoretic perspective. Consider i.i.d. training observations from an unknown population distribution $q_1$ over $\mathbb{R}^d$. A CNF is a time-dependent differomorphic map $\phi_t$ that transforms a random variable $x_0 \in \mathbb{R}^d$ from a source distribution $q_0$ into a random variable from $q_1$. The CNF induces a distribution of $x_t = \phi_t(x_0)$ at each time $t$, which is denoted by $p_t$, thereby forming a probability path $\{p_t : 0 \leq t \leq 1\}$. This probability path should (at least approximately) satisfy the boundary conditions $p_0 = q_0$ and $p_1 = q_1$. It is related to the flow map through the change-of-variable formula or the push-forward equation

$$p_t = [\phi_t]_* p_0.$$

FM aims at learning the corresponding vector field $u_t(x)$, which induces the probability path over time by satisfying the continuity equation (Villani, 2008).

The key observation underlying FM algorithms is that the vector field $u_t(x)$ can be written as a conditional expectation involving a conditional vector field $u_t(x|z)$, which induces a conditional probability path $p_t(\cdot|z)$ corresponding to the conditional distribution of $\phi_t(x)$ given $z$. Here, $z$ is the conditioning latent variable, which can be the target sample $x_1$ (e.g. Ho et al. (2020); Song et al. (2021); Lipman et al. (2023),) or a pair of $(x_0, x_1)$ on source and target distribution (e.g. Liu et al. (2023b); Tong et al. (2024a)). Specifically, Tong et al. (2024a), generalizing the result from Lipman et al. (2023), showed that

$$u_t(x) = \int u_t(x|z) \frac{p_t(x|z)q(z)}{p_t(x)} dz = \mathbb{E}\left(u_t(x|z)|x_t = x\right),$$

where the expectation is taken over $z$, which one can recognize is the conditional expectation of $u_t(x|z)$ conditional on the event that $x_t = x$. The integral is with respect to the conditional distribution of $z$ given $x_t = x$.

It is well-known in Bayesian decision theory (Berger, 1985) that under squared error loss, the Bayesian estimator, which minimizes both the posterior expected loss (which conditions on the data and integrates out the parameters) and the marginal loss (which integrates out both the parameters and the data), is exactly the posterior expectation of that parameter. This implies immediately that if one considers the conditional vector field $u_t(x|z)$ as the target of "estimation", and the corresponding "data" being the event that $x_t = x$, i.e., that the path goes through $x$ at time $t$, then the corresponding Bayes estimate for $u_t(x|z)$ will be exactly the marginal vector field $u_t(x)$, as it is now the "posterior mean" of $u_t(x|z)$. We emphasize again that here the "data" differs from the actual training and the generated noise observations, which in fact help form the "prior" distribution.

The FM algorithm is motivated from the goal of approximating the marginal vector field $u_t(x)$ through a smoother $v_t^\theta$ (typically a neural network), via the flow matching (FM) objective

$$\mathcal{L}_{\mathrm{FM}}(\theta) = \mathbb{E}_{t \sim U(0,1), x \sim p_t(x)} \|v_t^\theta(x) - u_t(x)\|^2,$$

which is not identifiable due to the non-uniqueness of the marginal vector fields that satisfy the boundary conditions without further constraints. In the following, we presume $t \sim U(0,1)$ and only show random variables to save notations. FM algorithms address this by fitting $v_t^\theta$ to the conditional vector field $u_t(x|z)$ after further specifying the distribution of $q(z)$ along with the conditional probability path $p_t(x|z)$, through minimizing the finite-sample version of the marginal squared error loss. This approach was referred to as the conditional flow matching (CFM) objective

$$\mathcal{L}_{\mathrm{CFM}}(\theta) = \mathbb{E}_{t, z \sim q(z), x \sim p_t(x|z)} \|v_t^\theta(x) - u_t(x|z)\|^2.$$

Traditionally, optimizing the CFM objective is justified because it has the same gradients w.r.t. $\theta$ to the corresponding FM loss (Lipman et al., 2023; Tong et al., 2024a). The Bayesian decision-theoretic perspective provides a further validation because approximating the conditional vector field by minimizing the marginal squared error loss can be interpreted as approximating the "posterior expectation" of $u_t(x|z)$, which is exactly $u_t(x)$.

Moreover, this is true for any coherently specified probability model $q(z)$. So long as the conditional probability path $p_t(x|z)$ is tractable, a suitable CFM algorithm can be designed. Therefore one can enrich the specification of $q(z)$ using Bayesian latent variable modeling strategies. This motivates us to generalize CFM training to the stream level, which we describe in the next section.

## 3 STREAM-LEVEL FLOW MATCHING

### 3.1 A PER-STREAM PERSPECTIVE ON FLOW MATCHING

A stream $\boldsymbol{s}$ is a stochastic process $\boldsymbol{s} = \{s_t : 0 \leq t \leq 1\}$, where each $s_t$ is a random variable in the sample space of the training data. We focus on streams connecting one end $x_0$ in the source to the other $x_1$ in the training data. From here on, $\boldsymbol{s}$ will take the space of the latent quantity $z$.

Instead of defining a conditional probability path and vector field given one endpoint at $t = 1$ (Lipman et al., 2023) or two endpoints at $t = 0$ and $1$ (Tong et al., 2024a), we shall consider defining it given the whole stream connecting the two ends. In order to achieve this, we need to specify a probability model for $\boldsymbol{s}$. This can be separated into two parts—the marginal model on the endpoints $\pi(x_0, x_1)$ and the conditional model for $\boldsymbol{s}$ given the two ends. That is

$$(x_0, x_1) \sim \pi \quad \text{and} \quad \boldsymbol{s}|s_0 = x_0, s_1 = x_1 \sim p_{\boldsymbol{s}}(\cdot|x_0, x_1).$$

Our model and algorithm will generally apply to any choice of $\pi$ that satisfies the boundary condition, including all of the examples considered in Tong et al. (2024a). We defer the description of specific choices of $p_{\boldsymbol{s}}(\cdot|x_0, x_1)$ to the next section and for now focus on the general framework.

Given a stream $\boldsymbol{s}$, the "per-stream" vector field $u_t(x|\boldsymbol{s})$ represents the "velocity" (or derivative) of the stream at time $t$, conditional on the event that $s_t = x$, i.e, the stream $\boldsymbol{s}$ passes through $x$ at time $t$. Assuming that the stream is differentiable with in time, the per-stream vector field is

$$u_t(x|\boldsymbol{s}) := \dot{s}_t = \mathrm{d}s_t/\mathrm{d}t,$$

which is defined only on all pairs of $(t, x)$ that satisfy $s_t = x$. The per-stream view extends previous CFM conditioning on endpoints and provides more flexibility. See Appendix A for more detailed discussion on how the per-stream perspective relates to the per-sample perspective on FM.

While the endpoint of the stream $s_1 = x_1$ is an actual observation in the training data, for the task of learning the marginal vector field $u_t(x)$, one can think of our "data" as the event that a stream $\boldsymbol{s}$ passes through a point $x$ at time $t$, that is $s_t = x$. Under the squared error loss, the Bayes estimate for the per-stream conditional vector field $u_t(x|\boldsymbol{s})$ will be the "posterior" expectation given the "data", which is exactly the marginal vector field

$$u_t(x) = \mathbb{E}(u_t(x|\boldsymbol{s})|s_t = x) = \mathbb{E}(\dot{s}_t|s_t = x). \tag{1}$$

Following Theorem 3.1 in Tong et al. (2024a), we can show that the marginal vector $u_t(x)$ indeed generates the probability path $p_t(x)$. (See the proof in the Appendix H.1.) The essence of the proof is to check the continuity equation for the (degenerate) conditional probability path $p_t(x|\boldsymbol{s})$.

A general stream-level CFM loss for learning $u_t(x)$ is then

$$\mathcal{L}_{\mathrm{sCFM}}(\theta) = \mathbb{E}_{t,\boldsymbol{s}}\|v_t^\theta(s_t) - u_t(x|\boldsymbol{s})\|^2 = \mathbb{E}_{t,\boldsymbol{s}}\|v_t^\theta(s_t) - \dot{s}_t\|^2$$

where the integration over $t$ is again $U(0,1)$ and that over $\boldsymbol{s}$ is with respect to the marginal distribution of $\boldsymbol{s}$ induced by $\pi(x_0, x_1)$ and $p_{\boldsymbol{s}}(\cdot|x_0, x_1)$. As in previous research such as Lipman et al. (2023); Tong et al. (2024a), we can show that the gradient of $\mathcal{L}_{\mathrm{sCFM}}$ equals that of $\mathcal{L}_{\mathrm{FM}}$ with details of proof in Appendix H.2. However, the stream-level CFM can be justified from a Bayesian decision theoretic perspective without gradient matching (Section 2). Because the (population-level) minimizer for the sCFM loss is $u_t(x)$, minimizing the sCFM loss provides a reasonable estimate for the marginal vector field $u_t(x)$. To see this, rewrite the sCFM loss by the law of iterated expectation as

$$\mathcal{L}_{\mathrm{sCFM}}(\theta) = \mathbb{E}_t\mathbb{E}_{\boldsymbol{s}}\left(\|v_t^\theta(s_t) - \dot{s}_t\|^2|t\right).$$

The inner expectation can be further written in terms of another iterated expection:

$$\mathbb{E}_{\boldsymbol{s}}\left(\|v_t^\theta(s_t) - \dot{s}_t\|^2|t\right) = \mathbb{E}_{s_t}\mathbb{E}_{\boldsymbol{s}}\left(\|v_t^\theta(s_t) - \dot{s}_t\|^2|t, s_t\right).$$

For any $x$, $\mathbb{E}_{\boldsymbol{s}}\left(\|v_t^\theta(s_t) - \dot{s}_t\|^2|\ t, s_t = x\right) = \mathbb{E}_{\boldsymbol{s}}\left(\|v_t^\theta(x) - \dot{s}_t\|^2|\ t, s_t = x\right)$, whose minimizer is the conditional expectation of $\dot{s}_t$ given $s_t = x$, which is exactly $u_t(x)$. Hence, one can estimate $u_t(x)$ by minimizing $\mathcal{L}_{\mathrm{sCFM}}(\theta)$. This justifies training $u_t(x)$ through the sCFM loss without regard to any specific optimization strategy.

## 3.2 Choice of the stream model

Next, we specify the conditional model for the stream given the endpoints $p_{\boldsymbol{s}}(\cdot|x_0, x_1)$. This model should emit streams differentiable with respect to time, with readily available velocity (either analytically or easily computable). Previous methods such as optimal transport (OT) conditional path (Liu et al., 2023b; Lipman et al., 2023; Tong et al., 2024a) can provide rather poor coverage of the $(t, x)$ space, resulting in extensive extrapolation of the estimated vector field $v_t(x)$. It is thus desirable to consider stochastic models for the streams that ensure the smoothness while allowing streams to diverge and provide more spread-out coverage of the $(t, x)$ space. Previous research suggested that, compared to ODEs, SDEs (Ho et al., 2020; Song et al., 2021) can be more robust in high-dimensional spaces (Tong et al., 2024b; Shi et al., 2023; Liu et al., 2023a), likely due to the robustness arising from the regularization induced by the additional stochasticity.

To preserve the "simulation-free" nature of CFM, we consider models where the joint distribution of the stream and its velocity is available in closed form. In particular, we further explore the streams following Gaussian processes (GPs). A desirable property of GP is that its velocity is also a GP, with mean and covariance directly derived from original GP (Rasmussen & Williams, 2005). This enables efficient joint sampling of $(s_t, \dot{s}_t)$ given observations from a GP in stream-level CFM training. By adjusting covariance kernels for the joint GP, one can fine-tune the variance level to control the level of regularization, thereby further improving the estimation of the marginal vector field $u_t(x)$ (Section 4.1). The prior path constraints can also be incorporated into the kernel design. Additionally, GP conditioning on the event that the stream passes through a finite number of intermediate locations between two endpoints again leads to a GP with analytic mean and covariance kernel (Section 4.2). This is particularly useful for incorporating multiple related observations.

Specifically, given $M$ time points $\boldsymbol{t} = (t_1, t_2, \ldots, t_M)$ with $t_1 = 0$ and $t_M = 1$, we let $\boldsymbol{s_t} = (s_{t_1}, s_{t_2}, \ldots, s_{t_M})$, and consider a more general conditional model for $p_{\boldsymbol{s}}(\cdot|\ \boldsymbol{s_t} = \boldsymbol{x}_{obs})$, where $\boldsymbol{x}_{obs} = (x_{t_1}, x_{t_2}, \ldots, x_{t_M})$ are a set of "observed values" that we require the statistic process $\boldsymbol{s}$ to

pass through at time $(t_1, t_2, \ldots, t_M)$. Note that this contains the special case of conditioning on two endpoints (i.e., $M = 2$) described in Section 3.1. We consider a more general construction for $M \geq 2$ because later we will use this to incorporate multiple related observations (such as time series or other measurements from the same subject).

We construct a conditional (multi-output) GP for $s$ that (approximately) satisfies the boundary conditions, with differentiable mean function $m$ and covariance kernel $k_{11}$. Since the derivative of a GP is also a GP, the joint distribution of $s$ and corresponding velocity process $\dot{s} := \{\dot{s}_t : t \in [0,1]\}$ given $s_t$ is also a GP, with the mean function for $\dot{s}$ be $\dot{m}(t) = \mathrm{d}m(t)/\mathrm{d}t$ and kernels defined by derivatives of $k_{11}$. To facilitate the construction of this GP, we consider an auxiliary GP on $s$ with differentiable mean function $\xi$ and covariance kernel $c_{11}$. Using the property that the conditional distribution of Gaussian remains Gaussian, we can obtain a joint GP model on $(s, \dot{s}) \mid s_t$, which satisfies the boundary conditions. For computational efficiency and ease of implementation, we assume independence of the GP across dimensions of $s$Notably, while we are modeling streams conditionally given $s_t$ as a GP, the marginal (i.e., unconditional) distribution of $s$ at all time points are allowed to be non-Gaussian, which is necessary for satisfying the boundary condition and for the needed flexibility to model complex distributions. The detailed derivation can be found in Appendix B, and the training algorithm for GP-CFM is summarized in Algorithm 1.

---

**Algorithm 1:** Gaussian Process Conditional Flow Matching (GP-CFM)

---

**Input** : observation distribution $\pi(\boldsymbol{x}_{\mathrm{obs}})$, initial network $v^\theta$, and a GP defining the conditional
distribution $(s_t, \dot{s}_t) \mid \boldsymbol{s_t} = \boldsymbol{x}_{\mathrm{obs}} \sim \mathcal{N}(\tilde{\boldsymbol{\mu}}_t, \tilde{\Sigma}_t)$, for $t \in [0,1]$.
**Output:** fitted vector field $v_t^\theta(x)$
**while** *Training* **do**
$\quad$ $\boldsymbol{x}_{\mathrm{obs}} \sim \pi(\boldsymbol{x}_{\mathrm{obs}}); t \sim U(0,1)$
$\quad$ $(s_t, \dot{s}_t) \mid \boldsymbol{s_t} = \boldsymbol{x}_{\mathrm{obs}} \sim \mathcal{N}(\tilde{\boldsymbol{\mu}}_t, \tilde{\Sigma}_t)$
$\quad$ $\mathcal{L}_{\mathrm{sCFM}}(\theta) \leftarrow \|v_t^\theta(s_t) - \dot{s}_t\|^2$
$\quad$ $\theta \leftarrow \mathrm{update}\,(\theta, \nabla_\theta \mathcal{L}_{\mathrm{sCFM}}(\theta))$
**end**

---

Several conditional probability paths considered in previous works are special cases of the general GP representation. For example, if we set $m(t) = tx_1 + (1-t)x_0$ (therefore, $\dot{m}(t) = x_1 - x_0$) and $k_{11}(t, t') = \sigma^2 \boldsymbol{I}_d$, the path reduces to the OT conditional path used in I-CFM with constant variance (Tong et al., 2024a). The I-CFM path can also be induced by conditional GP construction (Appendix B) using a linear kernel for $c_{11}$, with more details in Appendix C. In the following, we set $\xi(t) = 0$ and use squared exponential (SE) kernel for $c_{11}$ for each dimension (may with additional terms such as in Figure2). The details of SE kernel can be found in Appendix D.

Probability paths with time-varying variance, such as Song & Ermon (2019); Ho et al. (2020); Lipman et al. (2023), also motivate the adoption of non-stationary GPs whose covariance kernel could vary over $t$. For example, to encourage samples that display larger deviation from those in the training set (and hence more regularization), one could consider using a kernel producing larger variance as $t$ approaches to ends with finite training samples (Figure 2 and 7). Moreover, because the GP model for $s$ is specified given the two endpoints, both its mean and covariance kernel can be specified as functions of $(x_0, x_1)$. For example, if $x_1$ is an outlier of the training data, e.g., from a tail region of $q_1$, then one may incorporate a more variable covariance kernel for $p_{\boldsymbol{s}}(\cdot|x_0, x_1)$ to account for the uncertainty in the "optimal" transport path from $x_0$ to $x_1$.

## 4 NUMERICAL EXPERIMENTS

In this section, we demonstrate the benefits of GP stream models by several simulation examples. Specifically, we show that using GP stream models can improve the generated sample quality at a moderate cost of training time, through tweaking the variance function to reduce sampling variance of the estimated vector field. Moreover, the GP stream model makes it easy to integrate multiple related observations along the time scale.

### 4.1 Adjusting GP variance for high quality samples

We first show that one can reduce the variance in estimating $u_t(x)$ by incorporating additional stochasticity in the sampling of streams with appropriate GP kernels. As illustrated in Figure 1A, for estimating 2-Gaussian mixtures from standard Gaussian noise, the straight conditional stream used in I-CFM covers a relatively narrow region (gray). For points outside the searching region, there are no "data" and the neural network $v_t^\theta(x)$ must be extrapolated. In the sampling stage, this can lead to potential "leaky" or outlying samples that are far from the training observations.

For constructing GP conditional streams, we condition on the endpoints but expand the coverage region (red) by tweaking the kernel function (e.g. decrease the SE bandwidth in this case). This provides a layer of protection against extrapolation errors. We then train the I-CFM and GP-I-CFM 100 times using a 2-hidden layer multi-layer perceptron (MLP) with 100 training samples at $t = 1$, and calculate 2-Wasserstein (W2) distance between generated and test samples. For fair comparison, we set $\sigma = 0$ for I-CFM and use noise-free GP-I-CFM. The results are summarized in Table 1B. Empirically, the GP-I-CFM has smaller W2 distance than I-CFM. We further generate 1000 samples and streams for I-CFM and GP-I-CFM with largest W2 distance in Figure 1C, starting with the same points from standard Gaussian. In this example, several outliers are generated from I-CFM.

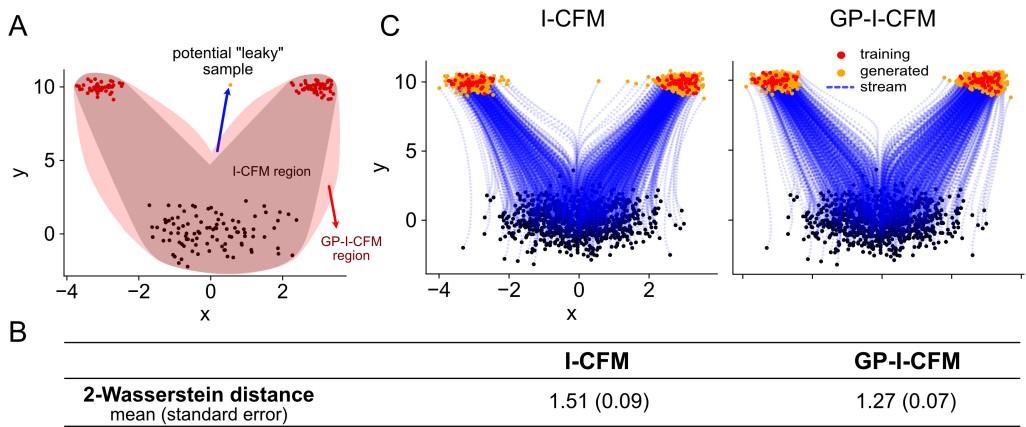

**B**

|  | I-CFM | GP-I-CFM |
|---|---|---|
| **2-Wasserstein distance** mean (standard error) | 1.51 (0.09) | 1.27 (0.07) |

Figure 1: **GP streams reduce extrapolation by expanding coverage area**. We use a 2-Gaussian mixture distribution as an example. Training observations are shown in red, generated samples in orange, and noise source samples in black. **A**. FM with straight conditional stream (e.g. I-CFM) may generate "leaky" or outlier samples due to extrapolation errors. The FM method with GP conditional stream has a broader coverage area. **B**. We train models with I-CFM and GP-I-CFM 100 times and calculate 2-Wasserstein (W2) distance between generated and test samples. Results of 100 seeds are summarized by mean and standard error. **C**. Among these 100 trained models, generate 1000 samples (orange) and streams (blue) for I-CFM and GP-I-CFM with largest W2 distance.

We can further modify the GP variance function over time to efficiently improve sample quality. Here, we consider the task of estimating and sampling from a 2-Gaussian mixture with 100 training samples at $t = 1$. For constant noise, diagonal white noise is added to perturb stream locations while retaining the SE kernel. For varying noise, we add a non-stationary dot product kernel to the SE kernel. Specifically, denote the kernel for auxiliary GP on $s$ in dimension $i$ as $c_{11}^i$, for $i = 1, \ldots, d$. Let $c_{11}^i(t, t') = c_{11}^{\text{SE}}(t, t') + \alpha t t'$ for increasing variance and $c_{11}^i(t, t') = c_{11}^{\text{SE}}(t, t') + \alpha(t-1)(t'-1)$ for decreasing variance, where $\{t, t'\} \in [0, 1]$ and $c_{11}^{\text{SE}}(t, t') = \sigma^2 \exp\left(-\frac{(t-t')^2}{2l^2}\right)$. (See Appendix B for additional details.) Some examples of the streams connecting two endpoints under different variance schemes are shown in Figure 2A. We train models 100 times and calculate 2-Wasserstein (W2) distance between generated and test samples, and the results of these 100 seeds are summarized in Table 2B. In this example, with infinite samples at $t = 0$ (from the standard Gaussian) but only 100 samples at $t = 1$, injecting noise at $t = 0$ worsens estimation. However, when approaching the target distribution ($t = 1$), adding noise can improve estimation with small samples (100). This noise perturbs the limited data, encouraging broader exploration and adding regularization to reduce estimation error.

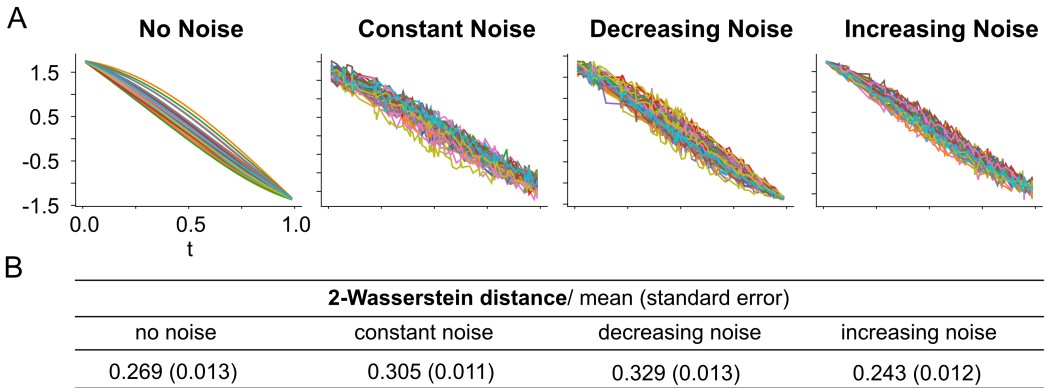

Figure 2: **Change variance over time by tweaking the covariance kernel**. We revisit the 2-Gaussian mixture distribution as an example.. **A**. Examples of conditional stream between two points, under different variance change scheme. **B**. We then train models under each variance scheme for 100 times and calculate 2-Wasserstein (W2) distance between generated and test samples for each. The results of 100 seeds are summarized by mean and standard error.

We further consider the transformation between two 2-Gaussian distributions with finite samples (100) at both ends. Results are shown in Appendix E. In this scenario, injecting noise near either endpoint improves estimation.

## 4.2 INCORPORATING MULTIPLE RELATED TRAINING OBSERVATIONS

Next, we show that GP streams enable the flexible inclusion of related observations along the same stream over time. This is particularly useful for generating related samples, such as time series (e.g., videos), where correlations between observations can enhance information sharing and improve estimation at each time point.

To illustrate the main idea, we consider 100 paired observations and place the two observations in each pair at $t = 0.5$ and $t = 1$ respectively (Figure 3A) while $t = 0$ still corresponds to a source distribution. Here, we show the generated samples (at $t = 0.5$ and $t = 1$) and the corresponding streams for GP-I-CFM and I-CFM. Again, 2-hidden layer MLP is used in this case. The I-CFM strategy employs two separate models with I-CFM algorithms (Figure 3B), whereas GP-I-CFM offers a single unifying model for all observations, resulting in a smooth stream across all time points (Figure 3C).

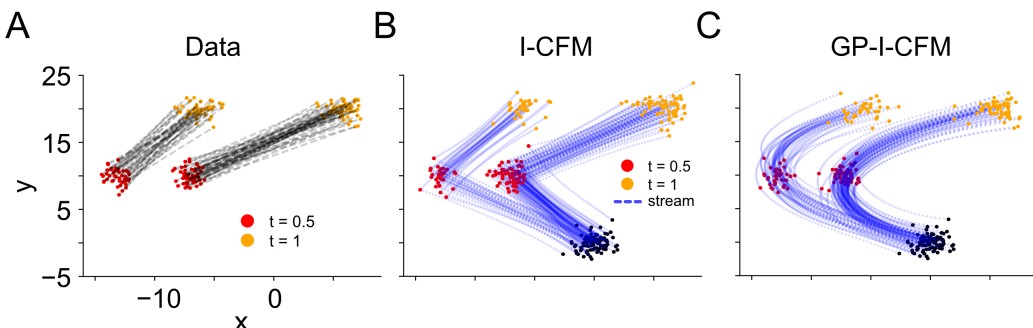

Figure 3: **GP streams can include related points flexibly**. **A**. Paired data with observations on t = 0.5 (red) and t = 1 (orange). **B**. The generated samples (red for t = 0.5 and orange for t = 1) and streams (blue) for I-CFMs. The I-CFMs contain two separate models trained by I-CFM, t = 0 (standard Gaussian noise) to t = 0.5 and t = 0.5 to t = 1. **C**. The generated samples for GP-I-CFM.

In some cases the GP streams may not be well separated and thus may confuse the training of the vector field at crossing points. In Figure 4, we show a time series dataset over 3 time points, where training data at $t = 0$ and $t = 1$ on one horizontal side while points at $t = 0.5$ are on the opposite side (Figure 4A). Therefore, these streams have two crossing regions (marked with blue boxes in Figure 4A), where the training of vector field is deteriorated when simply using the GP-I-CFM (Figure 4B). One easy solution is to further condition the neural net $v_t^\theta(x)$ on covariate (subject label) $c$, such that the optimizing objective is $\mathcal{L}_{\mathrm{cCFM}} = \mathrm{E}_{t\sim U(0,1), s\sim q(s|c)} \| v_t^\theta(s_t, c) - \dot{s}_t \|^2$, where $q(s \mid c)$ represents the distribution of $s$ given $c$. The validity for approximating the covariate-dependent vector field using the above optimizing objective is shown in the Appendix H.3. In this example, similar subjects have close starting points at $t = 0$, and we let $c = x_0$. By conditioning on $c$ (covariate model), the neural net are separated for different subjects, and hence the training of vector field will not be confused (Figure 4C).

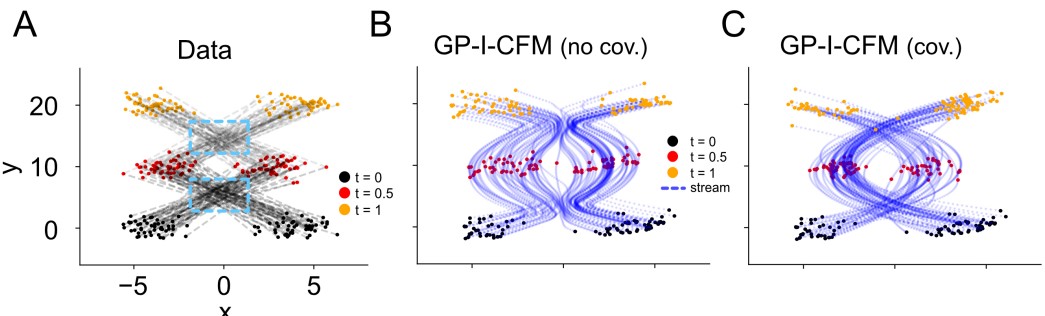

Figure 4: **Further conditioning on the starting points helps with stream generation**. **A**. Paired data with observations on three time points: t = 0 (black), t = 0.5 (red) and t = 1 (orange). The two stream cross regions are marked with light blue square. **B**. The generated samples and streams for GP-I-CFM (without covariate), where the initial points at $t = 0$ are generated from noise using a separate I-CFM. **C**. The generated samples and streams for GP-I-CFM with covariate using the same starting points, where the neural network is further conditioning on data at $t = 0$.

## 5 Applications

We apply our GP-based CFM methods to two hand-written image datasets (MNIST and HWD+), CIFAR-10 dataset and local field potential (LFP) dataset from mouse brain to illustrate how GP-based algorithms 1) reduce sampling variance (MNIST and CIFAR-10) and 2) flexibly incorporate multiple related observations (e.g. time series data) and generate smooth transformation across different time points (HWD+ and LFP dataset). The reported running times for the experiments are based on results obtained on a server configured with 2 CPUs, 24 GB RAM, and 2 RTXA5000 GPUs.

### 5.1 Variance Reduction

We explore the empirical benefits of variance reduction using FM with GP conditional streams on CIFAR-10 (Krizhevsky, 2009) the MNIST (Deng, 2012) database. For CIFAR-10, we compare performance for I-CFM and GP-I-CFM. Since OT strategy can be complementary to our GP-stream method to enhance the performance, we consider four algorithms in MNIST application: two linear stream models (I-CFM, OT-CFM) and two GP stream models (GP-I-CFM, GP-OT-CFM). Here, we show the details and results for CIFAR-10 application, the results for MNIST application (with more extensive experiments) can be found in the Appendix F.

We perform an experiment on unconditional CIFAR-10 generation (Krizhevsky, 2009) from a standard Gaussian source, using I-CFM and proposed GP-I-CFM, to evaluate the performance in the high-dimensional image setting. We use the similar setup to that of Tong et al. (2024a), such as time-dependent U-Net (Ronneberger et al., 2015; Nichol & Dhariwal, 2021) with 128 channels, a learning rate of $2 \times 10^{-4}$, clipping gradient norm to 1 and exponential moving average with a decay

of 0.9999. Besides, we add diagonal white noise $10^{-6}$ in GP-I-CFM, and set $\sigma = 10^{-3}$ in I-CFM for a fair comparison. The samples of $s_t$ is shown Figure 5B. The models are trained for 400,000 epochs, with batch size be 128. The I-CFM runs around 3.6 iterations per second, while GP-I-CFM runs around 3.0 iterations per second. The 64 generated images from I-CFM and GP-I-CFM are shown in Figure 5A, using a DOPRI5 adaptive solver. Visually, images generated by GP-I-CFM are generally sharper and exhibit more details compared to those generated by I-CFM (e.g. first row of Figure 5A). The Fréchet inception distance (FID) (Heusel et al., 2017), calculated by clean-fid library (Parmar et al., 2022) with 2000 samples, along training steps are plotted in Figure 5C, showing that the GP-I-CFM performs better than I-CFM. To show even a more significant performance improvement of GP-I-CFM, we may use a smaller GP bandwidth. However, this will lead to slower convergence, and it should be chosen by the practitioner to balance computational time (number of iterations) and sample quality.

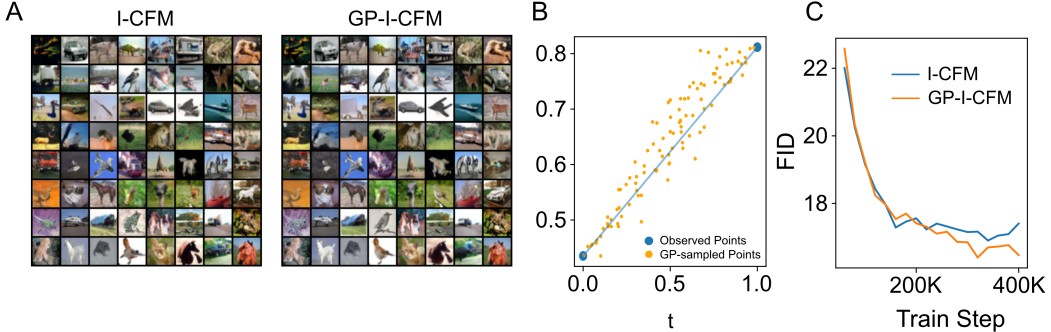

Figure 5: **Application to CIFAR-10 dataset**. Here, we compare the performance of unconditional image generation from standard Gaussian noise for I-CFM and GP-I-CFM. **A**. 64 generated samples for I-CFM and GP-I-CFM, starting from the same points. **B.** The samples of $x_t(\boldsymbol{s})$ from GP-I-CFM between a pair of two endpoints. **C.** FID of two algorithms over training steps.

## 5.2 MULTIPLE TRAINING OBSERVATIONS

Finally, we demonstrate how our GP stream-level CFM can flexibly incorporate related observations (between two endpoints at $t = 0$ and $t = 1$) into a single model and provide smooth transformation across different time points, using the HWD+ dataset (Beaulac & Rosenthal, 2022) and LFP dataset (Steinmetz et al., 2019), where LFP dataest is a time series data for mouse brain. Here, we show results for HWD+ dataset; refer to Appendix G for LFP application.

The HWD+ dataset contains images of handwritten digits along with writer IDs and characteristics, which are not available in MNIST dataset used in SectionF. Here, we consider the task of transforming from"0" (at $t = 0$) to "8" (at $t = 0.5$), and then to "6" (at $t = 1$). The intermediate image, "8", is placed at $t = 0.5$ (artificial time) for "symmetric" transformations. All three images have the same number of samples, totaling 1,358 samples (1,086 for training and 272 for testing) from 97 subjects. The U-Nets with 32 channels and 1 residual block are used. Both models with and without covariate (using starting images, as in Figure 4C) are considered. Each model is trained both by I-CFM and GP-I-CFM. The I-CFM transformation contains two separate models trained by I-CFM ("0" to "8" and "8" to "6"). Noise-free GP-I-CFM and I-CFM with $\sigma = 0$ are used for fair comparisons. In each training iteration, we randomly select samples within each writer, to preserve the grouping structure of data. The runtime for all algorithms (I-CFM, GP-I-CFM and corresponding labeled versions) are similar, which take 0.74s for passing all training data once. However, since I-CFMs contain 2 separated models, the running time is doubled.

The traces for 10 generated samples from each algorithm are shown in Figure 6A, where the starting images ("0" in the first rows) are generated by an I-CFM from standard Gaussian noise. Visually, the GP-based algorithms generate higher quality images and smoother transformation compared to algorithms using linear conditional stream (I-CFM), highlighting the benefit of including correlations across different time points. Additionally, the transformation generally looks smoother when the CFM training is further conditioned on the starting images.

We then quantify the performance of different algorithms by calculating the FID for "0", "8" and "6", and plot them over time for each (Figure 6B). For all FIDs, the GP-based algorithms (green & red) outperform their straight connection (I-) counterparts (blue & orange), especially for the FID for "8" at $t = 0.5$ and the FID to "6" at $t = 1$. This also holds for the FID for "0", as the GP-based algorithms are unified and the information is shared across all time points. This aligns with the observation by Albergo et al. (2023) that jointly learning multiple distributions better preserves the original image's characteristics during translation. However, for the I-algorithms, the conditional version (orange) performs worse than unconditional one (blue), as conditioning on the starting images makes the stream more separated, requiring more data to achieve comparable performance. In contrast, the data in GP-based algorithms is more efficiently utilized, as correlations across time points for the same subject are integrate into one model. Therefore, explicitly accounting for the grouping effect by conditioning on starting images (red) further improves performance.

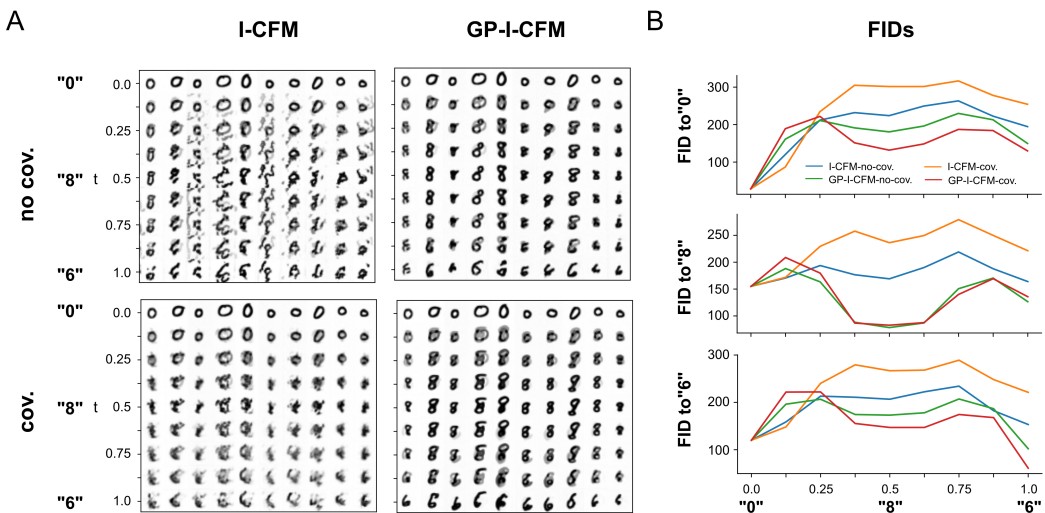

Figure 6: **Application to HWD+ dataset**. We fit models for transforming "0" to "8" and then to "6". Both covariate and non-covariate (on starting images) models are considered, and each model is fitted by both I-CFM and GP-I-CFM. The I-CFM transformation consists of two separate models trained by I-CFM ("0" to "8" and "8" to "6"). **A**. 10 sample traces for the four trained models. The starting images ("0"s in the first row) are generated by an I-CFM from standard Gaussian noise, and all four trained models use the same starting images. **B**. The corresponding FID to "0", "8" and "6" for these four trained models over time.

## 6 CONCLUSION

We have presented a Bayesian decision theoretic perspective to CFM training, which motivates an extension to CFM algorithms based on latent variable modeling. In particular, we adopt GP models on the latent streams. Our GP-CFM algorithm preserves the "simulation-free" feature of CFM training by exploiting distributional properties of GPs. This generalization not only reduces the sampling variance by expanding coverage of the sampling space in CFM training, but also allows easy integration of multiple related observations to achieve borrowing of strength.

There are some potential improvements either under GP-CFM frameworks or generally motivated by Bayesian decision theoretic perspective. For example, under GP-CFM framework, current implementations require the complete observations for all time points, which can be rare in time series applications. To deal with the missingness as well as the potential high-dimensionality of the training data, we may fit the GP-CFM in some latent space as in latent diffusion models (Rombach et al., 2022) and latent flow matching (Dao et al., 2023).

We believe that the Bayesian decision theoretic perspective and GP-CFM generalization so motivated open the door to various further improvements of CFM training of CNFs.

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

## A    DISCUSSION ON PER-STREAM PERSPECTIVE ON FLOW MATCHING

It is helpful to recognize the relationship between the per-stream vector field and the conditional vector field given one or both endpoints introduced previously in the literature. Specifically, the per-sample vector field in Lipman et al. (2023) corresponds to marginalizing out $s$ given the end point $x_1$, that is, $u_t(x \mid x_1) = \mathbb{E}\left(u_t(x \mid s) \mid s_t = x, s_1 = x_1\right)$. Similarly, the conditional vector field of Tong et al. (2024a), corresponds to marginalizing out $s$ given both $x_0$ and $x_1$, that is $u_t(x \mid x_0, x_1) = \mathbb{E}\left(u_t(x \mid s) \mid s_t = x, s_0 = x_0, s_1 = x_1\right)$. Furthermore, when $p_s(\cdot \mid x_0, x_1)$ is simply a unit-point mass (Dirac) concentrated on the optimal transport (OT) path, i.e., a straight line that connects two endpoints $x_0$ and $x_1$, then $u_t(x \mid s) = u_t(x \mid x_1) = u_t(x \mid x_0, x_1)$ for all $(s, t, x)$ tuples that satisfy $s_0 = x_0, s_1 = x_1, s_t = x$. Intuitively, when the stream connecting two ends is unique, conditioning on the two ends is equivalent to conditioning on the corresponding stream $s$. In this case, our stream-level FM algorithm (Section 3.2) coincides with those previous algorithms. More generally, however, this equivalence does not hold when $p_s(\cdot \mid x_0, x_1)$ is non-degenerate.

The per-stream view affords additional modeling flexibility and alleviates the practitioners from the burden of directly sampling from the conditional probability paths given one (Lipman et al., 2023) or both endpoints (Tong et al., 2024a). While the per-stream vector field induces a degenerate unit-point mass conditional probability path, we will attain non-degenerate marginal and conditional probability paths that satisfy the boundary conditions after marginalizing out the streams. Sampling the streams in essence provides a data-augmented Monte Carlo alternative to sampling directly from the conditional probability paths, which can then allow estimation of the marginal vector field $u_t(x)$ when direct sampling from the conditional probability path is challenging. Additionally, as we will demonstrate later, by approaching FM at the stream level, one could more readily incorporate prior knowledge or other external features into the design of the stream distribution $p_s(\cdot \mid x_0, x_1)$.

## B    DERIVATION OF JOINT CONDITIONAL MEAN AND COVARIANCE

For computational efficiency and ease of implementation, we assume independent GPs across dimensions and present the derivation dimension-wise throughout the Appendices. We use $s_t^i$ to denote the location of stream $s$ at time $t$ in dimension $i$, for $i = 1, \ldots, d$. Suppose each dimension of stream $s$ follows a Gaussian process with a differentiable mean function $\xi^i$ and covariance kernel $c_{11}^i$. Then, the joint distribution of $s_{t_1, \ldots, t_g}^i = (s_{t_1}^i, \ldots, s_{t_g}^i)'$ and $\dot{s}_{t_1, \ldots, t_g}^i = (\dot{s}_{t_1}^i, \ldots, \dot{s}_{t_g}^i)'$ at $g$ time points is

$$\begin{pmatrix} s_{t_1, \ldots, t_g}^i \\ \dot{s}_{t_1, \ldots, t_g}^i \end{pmatrix} \sim \mathcal{N}\left( \begin{pmatrix} \xi_{t_1, \ldots, t_g}^i \\ \dot{\xi}_{t_1, \ldots, t_g}^i \end{pmatrix}, \begin{pmatrix} \Sigma_{11}^i & \Sigma_{12}^i \\ \Sigma_{12}^{i\mathsf{T}} & \Sigma_{22}^i \end{pmatrix} \right), \tag{2}$$

where $\xi_t^i = \xi^i(t)$, $\dot{\xi}_t^i = \mathrm{d}\xi_t^i/\mathrm{d}t$, $\xi_{t_1, \ldots, t_g}^i = (\xi_{t_1}^i, \ldots, \xi_{t_g}^i)'$, $\dot{\xi}_{t_1, \ldots, t_g}^i = (\dot{\xi}_{t_1}^i, \ldots, \dot{\xi}_{t_g}^i)'$ and covariance $\Sigma_{jl}^i$ is determined by kernel $c_{jl}^i$. The kernel function for the covariance between $s$ and $\dot{s}$ in dimension

$i$ is $c_{12}^i(t,t') = \frac{\partial c_{11}^i(t,t')}{\partial t'}$, and the kernel defining covariance of $\dot{s}$ is $c_{22}^i = \frac{\partial^2 c_{11}^i(t,t')}{\partial t \partial t'}$ (Rasmussen & Williams (2005) Chapter 9.4). The conditional distribution of $(s, \dot{s})$ in dimension $i$ given $M$ observations $s_t^i = x_{\text{obs}}^i$ is also a (bivariate) Gaussian process. In particular, for $t \in [0,1]$, let $\boldsymbol{\mu}_t^i = (\xi_t^i, \dot{\xi}_t^i)'$ and $\boldsymbol{\mu}_{\text{obs}}^i = (\xi_{t_1}^i, \dots, \xi_{t_s}^i)$, the joint distribution is

$$\left(s_t^i, \dot{s}_t^i, {x_{\text{obs}}^i}'\right)' \sim \mathcal{N}\left(\begin{pmatrix} \boldsymbol{\mu}_t^i \\ \boldsymbol{\mu}_{\text{obs}}^i \end{pmatrix}, \begin{pmatrix} \Sigma_t^i & \Sigma_{t,\text{obs}}^i \\ {\Sigma^i}_{t,\text{obs}}^{\mathsf{T}} & \Sigma_{\text{obs}}^i \end{pmatrix}\right),$$

where $\Sigma_t^i = \text{Cov}(s_t^i, \dot{s}_t^i)$ and $\Sigma_{\text{obs}}^i = \text{Cov}(x_{\text{obs}}^i)$. Accordingly, the conditional distribution $(s_t^i, \dot{s}_t^i) \,|\, x_{\text{obs}}^i \sim \mathcal{N}(\tilde{\boldsymbol{\mu}}_t^i, \tilde{\Sigma}_t^i)$, where $\tilde{\boldsymbol{\mu}}_t^i = \boldsymbol{\mu}_t^i + \Sigma_{t,\text{obs}}^i {\Sigma_{\text{obs}}^i}^{-1}(x_{\text{obs}}^i - \boldsymbol{\mu}_{\text{obs}}^i)$ and $\tilde{\Sigma}_t^i = \Sigma_t^i - \Sigma_{t,\text{obs}}^i {\Sigma_{\text{obs}}^i}^{-1} {\Sigma^i}_{t,\text{obs}}^{\mathsf{T}}$.

## C  Optimal transport path from Conditional GP Construction

In this section, we show how to derive the path in I-CFM (Tong et al., 2024a) from the conditional GP construction (Appendix B) using a linear kernel. Without loss of generality, we present the derivation of "noise-free" path with $\sigma^2 = 0$ (i.e., the rectified flow, Liu et al. (2023b)).

Let $x_{\text{obs}}^i = (x_0^i, x_1^i)'$, $\xi_t^i = \dot{\xi}_t^i = 0$ and $c_{11}^i(t,t') = \sigma_a^2 + \sigma_b^2(t-1)(t'-1)$, such that

$$\Sigma_t^i = \begin{pmatrix} \sigma_a^2 + \sigma_b^2(t-1)^2 & \sigma_b^2(t-1) \\ \sigma_b^2(t-1) & \sigma_b^2 \end{pmatrix}, \qquad \Sigma_{t,\text{obs}}^i = \begin{pmatrix} \sigma_a^2 - \sigma_b^2(t-1) & \sigma_a^2 \\ -\sigma_b^2 & 0 \end{pmatrix},$$

$$\Sigma_{\text{obs}}^i = \begin{pmatrix} \sigma_a^2 + \sigma_b^2 & \sigma_a^2 \\ \sigma_a^2 & \sigma_a^2 \end{pmatrix}, \qquad\qquad {\Sigma_{\text{obs}}^i}^{-1} = \frac{1}{\sigma_b^2}\begin{pmatrix} 1 & -1 \\ -1 & 1 + \frac{\sigma_b^2}{\sigma_a^2} \end{pmatrix}.$$

Therefore,

$$\tilde{\boldsymbol{\mu}}_t^i = \Sigma_{t,\text{obs}}^i {\Sigma_{\text{obs}}^i}^{-1}\begin{pmatrix} x_0^i \\ x_1^i \end{pmatrix} = \begin{pmatrix} 1-t & t \\ -1 & 1 \end{pmatrix}\begin{pmatrix} x_0^i \\ x_1^i \end{pmatrix} = \begin{pmatrix} (1-t)x_0^i + tx_1^i \\ x_1^i - x_0^i \end{pmatrix},$$

$$\tilde{\Sigma}_t^i = \Sigma_t^i - \Sigma_{t,\text{obs}}^i {\Sigma_{\text{obs}}^i}^{-1} {\Sigma^i}_{t,\text{obs}}^{\mathsf{T}} = \boldsymbol{O}$$

## D  Covariance under Squared Exponential kernel

Throughout this paper, we adopted the squared exponential (SE) kernel, with the same hyperparameters for each dimension. The kernel defining block covariance for $s$, $(s, \dot{s})$ and $\dot{s}$ in dimension $i$ from Equation 2 are as follows:

$$c_{11}^i(t,t') = \alpha \exp\left(-\frac{(t-t')^2}{2l^2}\right) \qquad c_{12}^i(t,t') = \frac{\alpha}{l^2}(t-t')\exp\left(-\frac{(t-t')^2}{2l^2}\right)$$

$$c_{21}^i(t,t') = -c_{12}^i(t,t') \qquad c_{22}^i(t,t') = \frac{\alpha}{l^4}\left[l^2 - (t-t')^2\right]\exp\left(-\frac{(t-t')^2}{2l^2}\right).$$

## E  A Supplementary Example for Variance Changing over Time

Here, instead of generating data from standard Gaussian noise, we consider 100 training (unpaired) samples from a 2-Gaussian to another 2-Gaussian (Figure 7A). The example streams connecting two points under different variance schemes are shown in Figure 7B, again using additional nugget noise for constant noise, and a dot product kernel for decreasing and increasing noise, as described in Section 4.1. We then fit 100 independent models and calculate the W2 distance between generated and test samples at $t = 1$. The results are summarized in Figure 7C. Now, since both ends have finite samples, injecting noise (a.k.a. adding regularization) at both ends helps.

## F  Application to MNIST database

We explore the empirical benefits of variance reduction by using FM with GP conditional streams on the MNIST database (Deng, 2012). Four algorithms are considered: two linear stream models

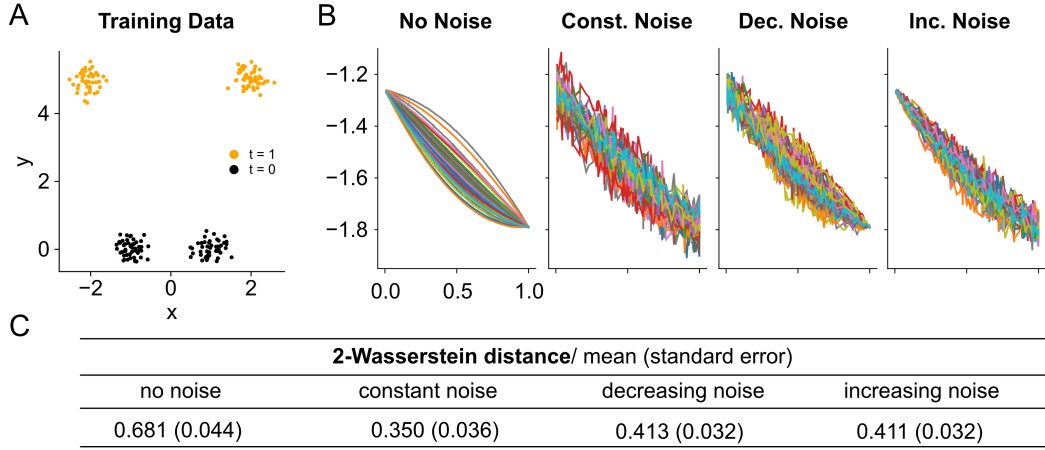

Figure 7: **Supplementary Example for Variance Change over Time**.**A**. The 100 observations in training data at $t = 0$ and $t = 1$. **B**. Examples of streams between two points, under different variance change scheme. **C**. Train models 100 times and calculate 2-Wasserstein (W2) distance between generated and test samples for each. The results of these 100 seeds are summarized by mean and standard error.

(I-CFM, OT-CFM) and two GP stream models (GP-I-CFM, GP-OT-CFM). For a fair comparison, we set $\sigma = 0$ for linear stream models and use noise-free GP stream models. For all models, U-Nets (Ronneberger et al., 2015; Nichol & Dhariwal, 2021) with 32 channels and 1 residual block are used. It takes around 50s, 51s, 52s and 53s for I-CFM, OT-I-CFM, GP-I-CFM and GP-OT-CFM to pass through all training dataset once. Figure 8A shows the 10 generated images for each trained model, starting from the same standard Gaussian noise. Compared to I-CFM, the OT version jointly samples two endpoints by 2-Wasserstein optimal transport (OT) map $\pi$ (Tong et al., 2024a). Here, we demonstrate how much the GP stream-level CFM can further improve the estimation. We train each algorithm 100 times, and calculate the kernel inception distance (KID) (Bińkowski et al., 2018) and Fréchet inception distance (FID) (Heusel et al., 2017). The histograms in Figure 8B show distribution of these 100 KIDs and FIDs, with results summarized in Figure 8C. According to KID and FID, the independent sampling algorithms (I-algorithms) are comparable to optimal transport sampling algorithms (OT-algorithms). However, algorithms using GP conditional stream exhibit lower standard error and fewer extreme values for KID and FID, thereby reducing the occurrence of outlier samples, as illustrated in Figure 1).

## G   APPLICATION TO LFP DATASET

In this section, to illustrate the usage of proposed GP-CFM for time series data, we apply the labeled-GP-I-CFM to a session of local field potential (LFP) data from a mouse brain. In the LFP dataset, the neural activity across multiple brain regions is recorded when the mice perform a task on choosing the side with highest contrast for visual gratings. The data contains 39 sessions from 10 mice, and each session contains multiple trials. Time bins for all measurements are 10 ms, starting 500 ms before stimulus onset. Here, we study LFP from stimulus onset to 500ms after stimulus, and hence each trial contains data from 50 time points. See Steinmetz et al. (2019) for more details of the LFP dataset.

Here, we choose recordings from a mouse in one session, where the trial is repeated 214 times. For each single trial, the data contains a time series from 7 brain regions. To illustrate the temporal smoothness over time in a visually significant way, we subset the data so that there are 5 evenly-spaced time points. In summary, the training data has 214 observations, and dimension for each observation is $5 \times 7$. Here, we fit the data by covariate GP-I-CFM, using the starting point as covariates, and generate 1000 LFP time series for each region (the starting LFP is generated from an I-CFM). For each second, the algorithm can run around 100 iterations per second (if we use all 50 time points, it runs around 2.5 iterations per second, and it take longer time to converge). The

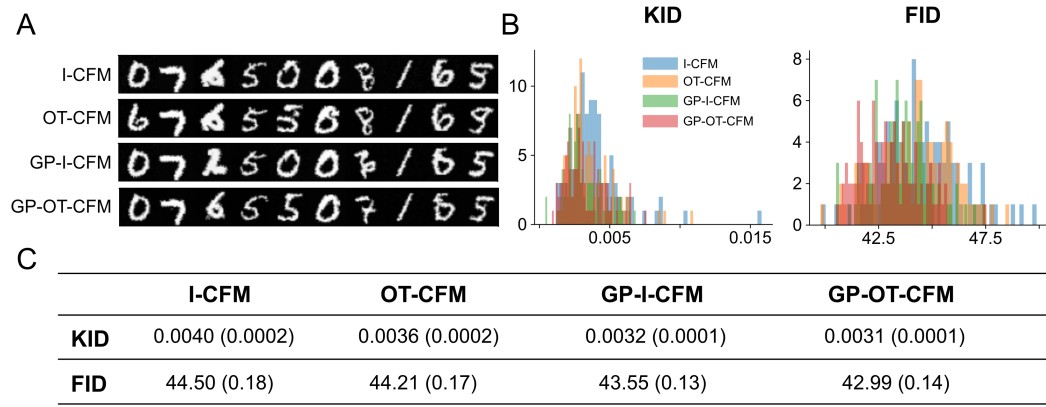

Figure 8: **Application to MNIST dataset**. We compare the performance of four algorithms (I-CFM, OT-CFM, GP-I-CFM and GP-OT-CFM) on fitting MNIST dataset. **A**. The 10 images generated from each trained model. Fit the models 100 times for each, and evaluate the quality of the samples by KID and FID. **B**. The histograms of KID and FID. **C**. The mean and standard error for KID and FID.

results are shown in Figure 9. The generated time series can further be used to study neural activity in different brain regions. For example, the mean trajectories in Figure 9A suggest that the LFPs in Cornu Ammonis region 3 (CA3) and dentate gyrus (DG) are highly correlated, which is consistent with the experiment fact that the rat DG does not project to any brain region other than the CA3 field of the hippocampus (Amaral et al., 2007). Besides this, we can use the generated samples to make more scientific insightful and concrete conclusions. But this is beyond the scope of this paper.

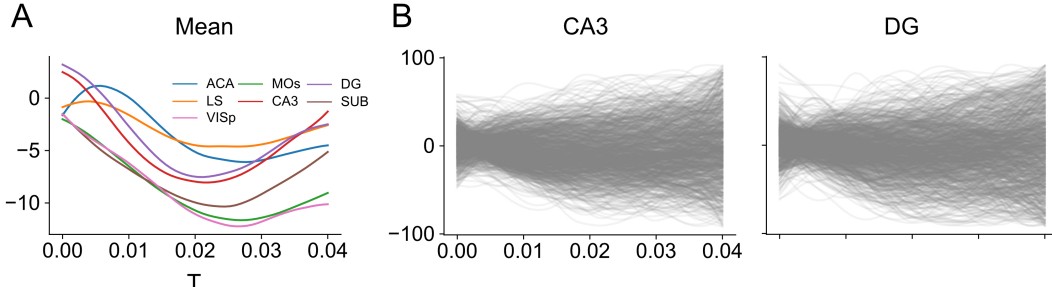

Figure 9: **Application to LFP data**.We apply the GP-I-CFM with covariate (on starting point) to a session of local field potential (LFP) data from 7 regions of mouse brain. In the training dataset, there are 214 observations (repeated trials). For each observation, it is a time series data of 5 time points from 7 brain regions. Here, we generate 1000 LFP time series for each region, where the starting LFP is generated from an I-CFM. **A.** The mean trajectories over 1000 samples. **B.** The generated 1000 time series for CA3 and DG.

## H    PROOF OF PROPOSITIONS

In this section, we provide proofs for several propositions in the main text. All these proofs are adapted from Lipman et al. (2023); Tong et al. (2024a).

### H.1    PROOF FOR CONDITIONAL FM ON STREAM

**Proposition 1.** *The marginal vector field over stream $u_t(x)$ generates the marginal probability path $p_t(x)$ from initial condition $p_0(x)$.*

*Proof.* Denote probability over stream as $q(\boldsymbol{s}) = \int p_{\boldsymbol{s}}(\boldsymbol{s} \mid x_0, x_1)\pi(x_0, x_1)d(x_0, x_1)$ and $p_t(x \mid \boldsymbol{s}) = \delta(x - s_t)$, then

$$\frac{d}{dt}p_t(x) = \frac{d}{dt}\int p_t(x \mid \boldsymbol{s})q(\boldsymbol{s})d\boldsymbol{s}$$

Assume the regularity condition holds, such that we can exchange limit and integral (and differentiation and integral) by dominated convergence theorem (DCT). Therefore,

$$= \int \frac{d}{dt}p_t(x \mid \boldsymbol{s})q(\boldsymbol{s})d\boldsymbol{s}$$

To handle the derivative on zero measure, define $s_t$-centered Gaussian conditional path and corresponding flow map as

$$p_{\sigma,t}(x \mid \boldsymbol{s}) := \mathcal{N}(x \mid s_t, \sigma^2 I)$$
$$\psi_{\sigma,t}(z \mid \boldsymbol{s}) := \sigma z + s_t,$$

for $z \sim N(0, I)$, such that $\lim_{\sigma \to 0} p_{\sigma,t}(x \mid \boldsymbol{s}) = p_t(x \mid \boldsymbol{s})$. Then by Theorem 3 of Lipman et al. (2023), the unique vector field defining $\psi_{\sigma,t}(z \mid \boldsymbol{s})$ (and hence generating $p_{\sigma,t}(x \mid \boldsymbol{s})$) is $u_t^*(x|\boldsymbol{s}) = ds_t/dt = u_t(s_t \mid \boldsymbol{s})$, for all $(t, x)$. Note that $u_t^*(x \mid \boldsymbol{s})$ extends $u_t(x \mid \boldsymbol{s})$ by defining on all $x$, and they are equivalent when $s_t = x$. Since $u_t^*(\cdot \mid \boldsymbol{s})$ generates $p_{\sigma,t}(\cdot \mid \boldsymbol{s})$, by continuity equation,

$$\frac{d}{dt}p_t(x) = \int \frac{d}{dt}\lim_{\sigma \to 0} p_{\sigma,t}(x \mid \boldsymbol{s})q(\boldsymbol{s})d\boldsymbol{s}$$
$$= \int -\lim_{\sigma \to 0}\text{div}(u_t^*(x \mid \boldsymbol{s})p_{\sigma,t}(x \mid \boldsymbol{s}))q(\boldsymbol{s})d\boldsymbol{s}$$

Then by DCT,

$$= -\lim_{\sigma \to 0}\text{div}\left(\int u_t^*(x \mid \boldsymbol{s})p_{\sigma,t}(x \mid \boldsymbol{s})q(\boldsymbol{s})d\boldsymbol{s}\right)$$
$$= -\text{div}\left(\int u_t^*(x \mid \boldsymbol{s})\lim_{\sigma \to 0}p_{\sigma,t}(x \mid \boldsymbol{s})q(\boldsymbol{s})d\boldsymbol{s}\right)$$
$$= -\text{div}\left(\mathbb{E}\left(u_t(x \mid \boldsymbol{s}) \mid s_t = x\right)p_t(x)\right)$$

By definition in equation 1,

$$= -\text{div}\left(u_t(x)p_t(x)\right),$$

which shows that $p_t(\cdot)$ and $u_t(\cdot)$ satisfy the continuity equation, and hence $u_t(x)$ generates $p_t(x)$. □

## H.2 PROOF FOR GRADIENT EQUIVALENCE ON STREAM

Recall

$$\mathcal{L}_{\text{FM}}(\theta) = \mathbb{E}_{t,x}\|v_t^\theta(x) - u_t(x)\|^2,$$
$$\mathcal{L}_{\text{sCFM}}(\theta) = \mathbb{E}_{t,\boldsymbol{s}}\|v_t^\theta(s_t) - u_t(x \mid \boldsymbol{s})\|^2,$$

where $x \sim p_t(x)$, $\boldsymbol{s} \sim q(\boldsymbol{s})$ and $q(\boldsymbol{s}) = \int p_{\boldsymbol{s}}(\boldsymbol{s} \mid x_0, x_1)\pi(x_0, x_1)d(x_0, x_1)$.

**Proposition 2.** $\nabla_\theta \mathcal{L}_{FM}(\theta) = \nabla_\theta \mathcal{L}_{sCFM}(\theta)$.

*Proof.* To ensure existence of all integrals and to allow the changes of integral (Fubini's Theorem), we assume that $q(\boldsymbol{s})$ are decreasing to zero at a sufficient speed as $\|\boldsymbol{s}\| \to \infty$ and that $u_t, v_t, \nabla_\theta v_t$ are bounded. To facilitate proof writing, let $p_t(x \mid \boldsymbol{s}) = \delta(x - s_t)$.

The L-2 error in the expectation ca be re-written as

$$\|v_t^\theta(x) - u_t(x)\|^2 = \|v_t^\theta(x)\|^2 + \|u_t(x)\|^2 - 2\langle v_t^\theta(x), u_t(x)\rangle$$
$$\|v_t^\theta(s_t) - u_t(x \mid \boldsymbol{s})\|^2 = \|v_t^\theta(s_t)\|^2 + \|u_t(x \mid \boldsymbol{s})\|^2 - 2\langle v_t^\theta(s_t), u_t(x \mid \boldsymbol{s})\rangle$$

Thus, it's sufficient to prove the result by showing the expectations of terms including $\theta$ are equivalent.

First,

$$\mathbb{E}_x \|v_t^\theta(x)\|^2 = \int \|v_t^\theta(x)\|^2 p_t(x) dx$$

$$= \int \int \|v_t^\theta(x)\|^2 p_t(x \mid \boldsymbol{s}) q(\boldsymbol{s}) dx d\boldsymbol{s}$$

$$= \mathbb{E}_{\boldsymbol{s}} \int \|v_t^\theta(x)\|^2 \delta(x - s_t) dx$$

$$= \mathbb{E}_{\boldsymbol{s}} \|v_t^\theta(s_t)\|^2$$

Second,

$$\mathbb{E}_x \langle v_t^\theta(x), u_t(x) \rangle = \int \langle v_t^\theta(x), u_t(x) \rangle p_t(x) dx$$

$$= \int \langle v_t^\theta(x), \frac{\int u_t(x \mid \boldsymbol{s}) p_t(x \mid \boldsymbol{s}) q(\boldsymbol{s}) d\boldsymbol{s}}{p_t(x)} \rangle p_t(x) dx$$

$$= \int \langle v_t^\theta(x), \int u_t(x \mid \boldsymbol{s}) p_t(x \mid \boldsymbol{s}) q(\boldsymbol{s}) d\boldsymbol{s} \rangle dx$$

$$= \int \int \langle v_t^\theta(x), u_t(x \mid \boldsymbol{s}) \rangle \delta(x - s_t) q(\boldsymbol{s}) d\boldsymbol{s} dx$$

$$= \mathbb{E}_{\boldsymbol{s}} \langle v_t^\theta(s_t), u_t(x \mid \boldsymbol{s}) \rangle$$

These two holds for all $t$, and hence $\nabla_\theta \mathcal{L}_{\text{FM}}(\theta) = \nabla_\theta \mathcal{L}_{\text{sCFM}}(\theta)$ □

### H.3 PROOF FOR GRADIENT EQUIVALENCE CONDITIONING ON COVARIATES

Let $x$ be response, $c$ be covariates, and $\boldsymbol{s}$ be the stream connecting two endpoints $(x_0, x_1)$. Given covariate $c$, denote the conditional distribution of $\boldsymbol{s}$ as $q(\boldsymbol{s} \mid c) = \int p_{\boldsymbol{s}}(\boldsymbol{s} \mid x_0, x_1, c) \pi(x_0, x_1) d(x_0, x_1)$ and marginal conditional probability path as $p_t(x \mid c)$. Further, let

$$\mathcal{L}_{\text{cFM}}(\theta) = \mathbb{E}_{t,x} \|v_t^\theta(x, c) - u_t(x \mid c)\|^2,$$

$$\mathcal{L}_{\text{cCFM}}(\theta) = \mathbb{E}_{t,\boldsymbol{s}} \|v_t^\theta(s_t, c) - u_t(x \mid \boldsymbol{s})\|^2,$$

where $x \sim p_t(x|c)$ and $\boldsymbol{s} \sim q(\boldsymbol{s} \mid c)$

**Proposition 3.** $\nabla_\theta \mathcal{L}_{cFM}(\theta) = \nabla_\theta \mathcal{L}_{cCFM}(\theta).$

*Proof.* To ensure existence of all integrals and to allow the changes of integral (Fubini's Theorem), we assume that $q(\cdot \mid c)$ decreases to zero at a sufficient speed as $\|\boldsymbol{s}\| \to \infty$ and that $v_t^\theta$, $u_t$, $\nabla_\theta v_t^\theta$ are bounded. To facilitate proof writing, let $p_t(x \mid \boldsymbol{s}) = \delta(x - s_t)$.

The L-2 error in the expectation ca be re-written as

$$\|v_t^\theta(x, c) - u_t(x \mid c)\|^2 = \|v_t^\theta(x, c)\|^2 + \|u_t(x \mid c)\|^2 - 2\langle v_t^\theta(x, c), u_t(x \mid c) \rangle$$

$$\|v_t^\theta(s_t, c) - u_t(x \mid \boldsymbol{s})\|^2 = \|v_t^\theta(s_t, c)\|^2 + \|u_t(x \mid \boldsymbol{s})\|^2 - 2\langle v_t^\theta(s_t, c), u_t(x \mid \boldsymbol{s}) \rangle$$

Thus, it's sufficient to prove the result by showing the expectations of terms including $\theta$ are equivalent.

First,

$$\mathbb{E}_x \|v_t^\theta(x, c)\|^2 = \int \|v_t^\theta(x, c)\|^2 p_t(x \mid c) dx$$

$$= \int \int \|v_t^\theta(x, c)\|^2 p_t(x \mid \boldsymbol{s}) q(\boldsymbol{s} \mid c) dx d\boldsymbol{s}$$

$$= \mathbb{E}_{\boldsymbol{s}} \int \|v_t^\theta(x, c)\|^2 \delta(x - s_t) dx$$

$$= \mathbb{E}_{\boldsymbol{s}} \|v_t^\theta(s_t, c)\|^2$$

Second,

$$
\begin{aligned}
\mathbb{E}_x \langle v_t^\theta(x,c), u_t(x \mid c) \rangle &= \int \langle v_t^\theta(x,c), u_t(x \mid c) \rangle p_t(x \mid c) dx \\
&= \int \langle v_t^\theta(x,c), \frac{\int u_t(x \mid \boldsymbol{s}) p_t(x \mid \boldsymbol{s}) q(\boldsymbol{s} \mid c) d\boldsymbol{s}}{p_t(x \mid c)} \rangle p_t(x \mid c) dx \\
&= \int \langle v_t^\theta(x,c), \int u_t(x \mid \boldsymbol{s}) p_t(x \mid \boldsymbol{s}) q(\boldsymbol{s} \mid c) d\boldsymbol{s} \rangle dx \\
&= \int \int \langle v_t^\theta(x,c), u_t(x \mid \boldsymbol{s}) \rangle \delta(x - s_t) q(\boldsymbol{s} \mid c) d\boldsymbol{s} dx \\
&= \mathbb{E}_{\boldsymbol{s}} \langle v_t^\theta(s_t,c), u_t(x \mid \boldsymbol{s}) \rangle
\end{aligned}
$$

These two holds for all $t$, and hence $\nabla_\theta \mathcal{L}_{\text{cFM}}(\theta) = \nabla_\theta \mathcal{L}_{\text{cCFM}}(\theta)$. $\qquad\square$

