# OpenReview forum: "Stream-level flow matching from a Bayesian decision theoretic perspective"
_ICLR.cc/2025/Conference — Submitted to ICLR 2025_

### Official Review · Reviewer_9wKq · 2024-10-28

**Soundness:** 2
**Presentation:** 3
**Contribution:** 2
**Rating:** 5
**Confidence:** 3

**Summary:**

This paper extends the Bayesian framework of flow matching, a generative algorithm that learns a vector field (parameterized by a neural network) that transforms noise to data. It extends conditional flow matching, which considers a target vector field conditioned on a data point, to have the target vector field conditioned on a stream, a complete path over the generation. It is shown that the minimizer of the resulting loss is the same as conditional flow matching. The streams are modeled by Gaussian Processes for analytical and computational convenience. Experiments are done on synthetic two-Gaussian datasets, MNIST, and HWD+ showing that this extension decreases the Wasserstein distance of the generations to the real data and enables control over the generative path.

**Strengths:**

- The paper is written clearly and straightforwardly.
- The proposed framework is theoretically sound, flexible, and novel. It shows that flow matching is general enough to be extended to different types of conditioning.
- Experiments show the value of the framework in a statistically significant way.

**Weaknesses:**

Overall, I feel that the paper does not quite fully back up some claims that it makes.
- The paper mentions that the computational cost is moderate, but does not explicitly discuss this in Section 3.2 or show measurements in Section 4. It would greatly strengthen the paper to include this, as GP posteriors can be expensive to compute.
- The HWD+ dataset is not usually used for time series, and so the construction of that experiment seems a bit synthetic.

**Questions:**

- Is the assumption of independence among the dimensions in the GP too restrictive? Does it allow generating data with arbitrary dependence structures?
- Have the authors done experiments on classic time series datasets, e.g. any of the UCR time series datasets (https://www.cs.ucr.edu/~eamonn/time_series_data_2018/)?

---

> ### Author Response · Authors · 2024-11-22
> **Response to Reviewer 9wKq**
>
> 1. > The paper mentions that the computational cost is moderate, but does not explicitly discuss this
>
> In the updated manuscript, we have included computational time for the applications. The empirical cost is almost identical to that of I-CFM and OT-CFM. Since we assume independent GPs for each dimension, the data's dimensionality does not pose a significant issue (e.g., in the CIFAR-10 example discussed in General Comments 1, GP-I-CFM is only slightly slower than I-CFM). For two endpoints or data with multiple time points, matrix inversion for the GP is computationally inexpensive.
>
> However, for very long time-series data, some strategies (e.g., sparse GP approximations) may be needed to reduce matrix inversion costs. Such considerations are beyond the scope of this paper.
>
> 2. > The HWD+ dataset is not usually used for time series, and so the construction of that experiment seems a bit synthetic.
>
> The HWD+ example demonstrates how GP can capture within-subject correlations (grouping effects) across all observations. While our stream-based approach place related observations along time from $t=0$ and $t=1$, the main purpose is to share information across the related samples and such related samples do not have to actually come from a time-series. See also our responses to Reviewer 1 regarding the discussion on modeling correlation through random effects of the grouping structure. Additionally, we have included an LFP example to illustrate the use of GP-CFM for an actual time-series data. For details, refer to General Comments 2 and Appendix G in the updated manuscript.
>
> 3. Is the assumption of independence among the dimensions in the GP too restrictive?
>
> The GP model on the stream is used to design the conditional path, facilitating sample generation. It does not affect the correlation structure of the endpoints. Using a dependent GP could potentially improve sampling efficiency by explicitly modeling the correlation of the path across different dimensions, rather than relying on implicit capture via a unified neural network for the vector field. However, this approach would be computationally expensive and impractical for high-dimensional settings. Furthermore, I-CFM and other FM models assume independent paths for each dimension, and our GP-CFM builds on and extends these methods.

---

> > ### Comment · Reviewer_9wKq · 2024-11-27
> >
> > Thank you for the rebuttal and the updated manuscript. After reading them and the other reviews, I have decided to keep my score.

---

### Official Review · Reviewer_SUwj · 2024-11-03

**Soundness:** 3
**Presentation:** 2
**Contribution:** 2
**Rating:** 3
**Confidence:** 3

**Summary:**

The paper proposes to use Gaussian Process (GP) models to define interpolants for flow matching. This is motivated by first introducing a “Bayesian decision theoretic perspective” on flow matching, essentially corresponding to the observation that the marginal flow can be obtained as a posterior expectation of a conditional flow, which in turn is the solution to a least squares regression problem. The paper also considers the case of conditioning the GP interpolant on intermediate values of the flow (that is, if the flow transports q_0 to q_1 over the time interval t \in [0,1], then we can condition on observations of the trajectories at, say, t=0.5).

**Strengths:**

Using GPs to define stochastic interpolants gives a flexible framework that makes use of several convenient properties of GPs, specifically that the derivative of a GP is also a GP (used in the definition of the objective function, since we regress the learned flow on the velocity field induced by the interpolant), and the fact that we can compute a conditional GP by conditioning on an arbitrary collection of points (used to enable conditioning on intermediate samples, e.g. at t=0.5). It also opens up for using different tools and techniques from the GP regression literature in the context of flow-based generative models.

**Weaknesses:**

The novelty of the work is questionable.
* As far as I can tell, the “Bayesian perspective” developed in Section 2, as well as the “per-stream perspective” develop in Section 3.1 is known and covered in prior work, e.g. https://arxiv.org/abs/2303.08797
* The use of “GP regression models” to define the interpolant is (to the best of my knowledge) novel, but at the same time quite similar to using SDEs, see e.g. Section 3.1 in https://arxiv.org/abs/2303.08797 and methods based on Schrödinger bridges, e.g. https://arxiv.org/abs/2303.16852 (the latter is cited in the current paper, but only in passing and no discussion about similarities with this work is provided). Note that these models based on “diffusive interpolants” also result in GPs, although defined indirectly through the corresponding SDE. While not identical to the current paper (which is based on directly defining the GP through mean and covariance functions) I would have liked to see an in-depth discussion about similarities and differences/pros and cons.

Based on this, I would recommend rewriting the paper and move section 2 and 3.1 into a background section (relating these to prior work) and instead elaborating on the (as far as I can tell) novel contribution of this work, namely the use of GPs to define interpolants. You could for instance elaborate on the difference between k11 and c11 (which is not so clear in the current paper), how you define the unconditional GP marginals for the endpoints x0, x1 (I assume that you use an unconditional reference GP which is then conditioned on the non-Gaussian data points to obtain the interpolant?), and pros and cons between the GP-based model compared to other diffusive interpolants/bridges.

**Questions:**

The possibility of conditioning on intermediate states is interesting, but I wonder how this is related to multi-marginal optimal transport, e.g. https://arxiv.org/abs/2310.03695 How does the proposed method differ from the multi-marginal setting?

A minor comment is that, in my opinion, using all of z, x_t, and s to refer to more of less the same thing is unnecessary and the paper would be easier to read if you harmonize the notation.

---

> ### Author Response · Authors · 2024-11-22
> **Response to Reviewer SUwj**
>
> Thanks a lot to the reviewer for their constructive comments, and here we clarified some specific points…
>
>
> 1. > As far as I can tell, the “Bayesian perspective” developed in Section 2 (of https://arxiv.org/abs/2303.08797).
>
> Section 2 of the referenced paper establishes the stochastic interpolant framework, which approaches FM model construction from a hierarchical Bayesian perspective. However, the "Bayesian decision-theoretic perspective" referenced here specifically refers to the justification of CFM training.
>
> Specifically, we show that one can justify the claim that optimizing the FM objective is equivalent to optimizing the CFM objective using Bayesian decision theory. This justification does not rely on any gradient-related assumptions, and thus, under this perspective, we are not restricted to gradient descent for parameter estimation. Additionally, viewing parameter estimation from this perspective allows for broader generalizations of the FM method, with stream-level FM being one such example. For a more detailed discussion, see Section 2.
>
> 2. > the “per-stream perspective” develop in Section 3.1 is known and covered in prior work (https://arxiv.org/abs/2303.08797).
>
> Section 3.1 of the referenced paper defines the diffusive interpolant, which conditions on two endpoints while making the path stochastic using a Brownian bridge. In contrast, we condition on the entire stream, which can be either stochastic or deterministic. While the per-stream vector field induces a degenerate unit-point mass conditional probability path, marginalizing over the streams yields non-degenerate marginal and conditional probability paths that satisfy the boundary conditions. **The Brownian bridge used in the referenced paper is a special case of a Gaussian process (GP)**, and therefore, conditioning on the entire stream and modeling it using a GP offers greater modeling flexibility. For more details, see the discussion in Appendix A.
>
> 3. > Note that these models based on “diffusive interpolants” also result in GPs, although defined indirectly through the corresponding SDE. While not identical to the current paper, I would have liked to see an in-depth discussion about similarities and differences/pros and cons.
>
> Using the GP-stream, we can explicitly and flexibly design the path through the mean and covariance functions. Compared to a GP induced by an SDE, this approach is more 1) flexible (Brownian bridge is a special case of GP), 2) more straightforward for incorporating prior constraints on the path (e.g., periodic patterns in time-series data) and 3) accommodating multiple observations.
>
> 4. > The possibility of conditioning on intermediate states is interesting, but I wonder how this is related to multi-marginal optimal transport, e.g. https://arxiv.org/abs/2310.03695
>
> Interesting paper! The mentioned multi-marginal OT (MMOT) paper learns joint distributions by generalizing the stochastic interpolant. Both GP-stream and MMOT can capture correlations among different marginal distributions, but GP-stream offers greater flexibility for path design. This flexibility is particularly advantageous in scenarios with path constraints or prior knowledge (e.g., periodic time-series data). Nonetheless, we appreciate the reviewer bringing this paper to our attention. The benefits observed in their work align with our findings in the HWD+ data application.
>
>
> 5. > A minor comment is that, in my opinion, using all of z, x_t, and s to refer to more of less the same thing is unnecessary and the paper would be easier to read if you harmonize the notation.
>
> Thanks for the suggestion! In the updated manuscript, we have changed our notation throughout and got rid of the $x_t(s)$ and $\dot{x}_t(s)$ notation completely, which indeed makes the math much cleaner and easier to follow.

---

> > ### Comment · Reviewer_SUwj · 2024-12-02
> >
> > I thank the authors for their reply. I have read the rebuttal and updated paper, and I still believe that the novelty in the "Bayesian perspective" and "per-stream perspective" is not sufficiently novel to warrant publication. As mentioned in my review, the used of GPs in this context is interesting, and my recommendation would be to focus more on the flexibility offered by this approach when revising the paper, both in the method description and numerical evaluation.

---

### Official Review · Reviewer_yJK9 · 2024-11-04

**Soundness:** 2
**Presentation:** 3
**Contribution:** 2
**Rating:** 3
**Confidence:** 3

**Summary:**

The paper presents an informal view of the CFM framework from the perspective of the Bayesian approach. A new vector field learning algorithm (GP-CFM) is presented, which leads to more intuitively expected trajectories and slightly improves the metrics in simple cases, compared to the classical I-CFM and OT-CFM.

**Strengths:**

- Python code is provided
- The paper is rather clear written

**Weaknesses:**

- There are no theoretical estimates, such as the accuracy of the presented Algorithm, rate of the convergence, variance reduction etc. Thus, the paper is empirical in essence.
- Experiments have been performed only on synthetic data or low-dimensional datasets like MNIST. I would like to see more experiments including experiments on higher dimensional data, especially since the paper is empirical.
The results on CIFAR10 that Tong et al 2024b cite would be much more convincing.
The classic paper by Lipman et al “Flow Matching for Generative Modeling”, 2023, considers an ImageNet dataset up to $128\times128$. I recommend present the results also on ImageNet at different dimensions.
- In the same way, I suggest publishing other metrics for comparison, such as NLL and NFE (as in Lipman et al).
- As the results of Fig. 5 show, the difference in metrics between the presented method and I-CFM (or OT-CFM) is extremely small. Thus, the method does not show a significant gain in efficiency. I would expect a gain of at least 10%--15% or more. However, as I wrote in the previous point, it would be much more convincing to experiment on multiple high-dimensional datasets rather than two similar ones.
- If I am not mistaken, the only significant difference between the presented algorithm and classical CFM is the presence of empirical points within the trajectory (i.e., points for time $t$ other than 0 or 1). However, this formulation goes beyond the original formulation -- if we want to sample from an unknown distribution given by a set of samples, we have no interior points in the trajectory, only a set of endpoints. Moreover, different FM models may use different maps, and hence different intermediate points for given source and target distributions. Could you please elaborate how your method handle the setting with only two (end-)points and how your implementation takes into account different maps?
- In [1] there is an example of an experiment with points lying inside the trajectory. It would be interesting to see the application of the presented method to this data, since such a problem statement falls exactly under the presented framework. This is a recent article, and it just discusses time-dependent processes (of cell evolution).
- In Fig. 1 C, I-CFM appears to be simply undertrained. Can you elaborate on the details of the experiments and how calculate the metric in Table on this Fig? Can you present another metrics, like NLL, in this case? Am I correct that the pictures on C show the worst results among 100 runs, not the typical results?
- The theoretical part of the paper is written in a way that is not rigorous and sloppy in terms of mathematics (although it claims some theoretical results). These include:
  - It is not always clear for which variables the mathematical expectation is taken.
     - For example, L173--174: $\mathbb{E}_{t,q(s),\delta(x-x_t(s))}$. The first variable on which integration is taken is $t$ (and we know from the context that it is uniformly distributed in the interval $[0,1]$). But then there is $q(s)$ under the sign $\mathbb E$. This is already a distribution function (or is it? It has a fixed argument), not a variable. Such confusion leads to heavy reading and increases the possibility of errors in the text itself. Further $,\delta(x-x_t(s))$ is under $\mathbb E$, where there are already several variables under the function sign.  I advise (at least the first time you encounter a formula) to write clearly on which variable the integral is taken and what the measure (or probability density function) of this random variable is, or better yet, to write all the key mathematical expectations that result from this paper in the form of integrals (Riemann or Lebesgue) in the Appendix.
     - As a corollary to the previous point. In probability theory and mathematical statistics, mathematical expectation is nothing but an integral over a measure. In the paper, L185--186, what is meant by $\mathbb E_{x_t(s)}$, over which variable (measure) is the integration performed? If this variable has a distribution density function, could you give it explicitly?
  - L141-142: what is the strict definition of $s$?  Is $s$ is just a curve: $s\colon [0,1]\to \mathbb R^d$, $s(0)=x_0$, $s(1)=x_1$? Or is it stochastic process https://en.wikipedia.org/wiki/Stochastic_process? (For example, a Brownian bridge is a continuous-time gaussian process with the given start and end points). In both cases it is customary to write $s(t)$ (or $s_t$), not $x_t(s)$. I suggest to clarify definitions, and use the standardized designations adopted for these definitions. The same applies to the definition of so-called ``stream velocity'',  L155-156. Namely, if $X(t)$ is stochastic process, then $dX_t = \mu_t\ dt + \sigma_t\ dB_t$, where $B(t)$ is Wiener process and the concept of a derivative has to be further defined. Thus, the notation $dX(t)/dt$ is ambiguous. Please clarify these concepts that are key to your paper.
  - It is not a strict designation just a vertical line, such as L150--151, or L239-240. Does it denote the conditional expectation (https://en.wikipedia.org/wiki/Conditional_expectation)? I suggest to use strict mathematical notation (such as conditional expectation) to avoid misunderstandings, at least in places where the essence of the presentation is theoretical exploration.
  - Usually, the symbol $\mathcal N(\mu, \sigma)$ refers to the (multivariate) Gaussian distribution. How then to **strictly** understand sampling of both $x_t$ and its derivative simultaneously in L239-240 (from a single Gaussian distribution with time-dependent parameters)? How is the conditional expectation on $x_\text{obs}$ taken during sampling? Can you please give a detailed step-by-step description of the calculations, in particular, how exactly you sample $x_t$  and  $\dot x_t$, how conditional expectation is taken (if it was taken)?

 - L073-074 ` to integrate related observations`. It is not clear what are `related` observations in general. Can you clarify what these are and in which real-world problem statements they occur? Further on you indicate that for MNIST dataset related are pictures with numbers are 0, 8, 6. This seems like an _ad hoc_ statement. How can you extend this concept (of `related` observations) to, for example, CIFAR10 dataset and clarify how can it help in the solving initial task of sampling from unknown distribution?

 - L472--474 `Here, we consider the task of transforming from“0” (at t = 0) to “8” (at t = 0.5), and then to “6” (at t = 1).` Such a formulation seems too artificial (and beyond the scope of the classical FM problem formulation). Can you please explain the significance of this formulation, and/or cite similar real-world problems that include such a formulation, and/or papers with such a problem formulation?



[1] Alexander Y. Tong, Nikolay Malkin, Kilian Fatras, Lazar Atanackovic, Yanlei Zhang, Guillaume Huguet, Guy Wolf, and Yoshua Bengio. Simulation-free Schr¨odinger bridges via score and flow matching. In Sanjoy Dasgupta, Stephan Mandt, and Yingzhen Li (eds.), Proceedings of The 27th International Conference on Artificial Intelligence and Statistics, volume 238 of Proceedings of Machine Learning Research, pp. 1279–1287. PMLR, 02–04 May 2024b.

**Questions:**

In addition to the questions in the weakness section, please answer the following questions:

- L247--249: What is the relationship between your method and OT-CFM? How does sample pairing, which is explicitly present in OT-CFM (Tong et al, 2024) in the form of OT-minibactch, arise in your method? Moreover, OT-minibatch can use different metrics to find the distance between points, which one do you use?

- Can you explain in a little more detail how you found the FID for MNIST? In the zip archive there is a file metric/Fid_score.py, there basic functions require an image size of $299\times299$ (e.g. line 53 of this file: `assert x.shape[1:] == (3, 299, 299), “Expected input shape to be: (N,3,299,299,299)” +\`). However, in line 186 resize is commented out: `#im = cv2.resize(im, (299, 299))`, and the images from MNIST are $28\times28$ in size.

- On L071--072 `We demonstrate that adjusting the GP streams can reduce the variance of the estimated marginal vector field with moderate computational cost.` Can you explain in more detail where in the article you showed this and in what form (theoretically, empirically)?

---

> ### Author Response · Authors · 2024-11-22
> **Response to Reviewer yJK9 (Part 1)**
>
> Thanks a lot to the reviewer for their constructive comments. Here, we clarified some specific points…
>
> 1. > I would like to see more experiments including experiments on higher dimensional data
>
> We have now included a CIFAR-10 example. See the detailed discussion in General Comments 2 and the updated manuscript in Section 5.1.
>
>
> 2. > As the results of Fig. 5 show, the difference in metrics between the presented method and I-CFM (or OT-CFM) is extremely small.
>
> In the low-dimensional case, since I-CFM performs well enough, there is limited room for GP-I-CFM to demonstrate significant improvement. We have added a high-dimensional example using CIFAR-10, as described in General Comments 2 and Section 5.1. In general, the extent of improvement depends on the GP tuning parameter. Introducing more stochasticity in the stream allows the algorithm to explore a wider space (as illustrated in Figure 1A) during training, thereby reducing sampling variance. However, this also slows down convergence. Thus, practitioners should select the GP tuning parameter to balance computational cost and sample quality.
>
> 3. > If I am not mistaken, the only significant difference between the presented algorithm and classical CFM is the presence of empirical points within the trajectory (i.e., points for time other than 0 or 1).
>
> No, that's not the main difference. The main difference is in the way we sample points between 0 and 1, whether or not those points are empirical or not. A side product of our approach is that it offers the additional ability to include multiple empirical observations ($\geq 2$) along the sampled GP path but that is not the key defining characteristic of our approach. Compared to I-CFM, instead of using linear interpolation (so that all sampled points are along the straight line connecting $x_0$ and $x_1$), we model and thus sample the streams connecting two endpoints from a GP, which has several benefits: 1) the stream connecting two endpoints is now stochastic, and hence the algorithm can search for wider region to reduce the sampling variance. 2) It can easily incorporate constraint/ prior of path into GP mean and kernel design. 3) It can easily incorporate multiple time points as mentioned by the reviewer (2 endpoints case is special for this).
>
> 4. > different FM models may use different maps, and hence different intermediate points for given source and target distributions. Could you please elaborate how your method handle the setting with only two (end-)points and how your implementation takes into account different maps?
>
> The "intermediate points" we consider here are observations (e.g., different time points in a time series), so they remain consistent across different FM models. Observing only two endpoints is a special---in fact the baseline---case of our method. When we only condition on $x_0$ and $x_1$, we sample multiple streams connecting $x_0$ and $x_1$ along with the corresponding velocity process from the corresponding GP.
>
> 5. >  It would be interesting to see the application of the presented method to this data, since such a problem statement falls exactly under the presented framework.
>
> The "baseline" case without any "intermediate observations" and conditions only on the two endpoints is the one applied in most of our simulations (e.g., Section 4.1) and applications (e.g., CIFAR-10 and MNIST).
> Beyond these examples, we demonstrate the more general case that conditions on more than the two endpoints in Section 4.2 and Section 5.2. The HWD+ dataset application includes points lying within the trajectory. In this revision, we also added the new LFP application example to illustrate how our method can be applied to time-series data (see details in General Comments 3 and the updated manuscript in Appendix G).
>
> 6. > In Fig. 1 C, I-CFM appears to be simply undertrained.
>
> We do not believe the phenomenon is due to an undertraining of I-CFM. For I-CFM, we have used a 3-hidden-layer MLP with a hidden dimension of 64. The learning rate is set to $1e-3$, and the model is trained for 10,000 epochs, which should be sufficient for such a simple example. Instead, we believe it is due to overfitting of I-CFM due to extrapolation of the MLP in the sparsely sampled regions. One can potentially enforce stronger relugarization to reduce such overfitting and therefore improve the fit of I-CFM. Our apporach, by increasing the sampling region through GP paths, can be considered one of such approaches.

---

> ### Author Response · Authors · 2024-11-22
> **Response to Reviewer yJK9 (Part 2)**
>
> 7. > The theoretical part of the paper is written in a way that is not rigorous and sloppy.
>
> Thank you for the suggestions. They are extremely helpful. We have updated our notations and rewritten the mathematical expressions and we believe the new version is substantially clearer than our original submission. The major changes have beensummarized in General Comments 1. Here are some additional clarifications
>
> - The subscripts of the expectation are now random variables.
>
> - We have redefined the stream $s$ and replaced the previously used $x_t(s)$ with $s_t$, a random variable denoting the location of the stream $s$ at time $t$.
>
> - We now use $\mid$ for conditional expectations, rather than simple vertical lines.
>
> - For joint sampling of $(s_t, \dot{s}_t)$, since derivative is a linear operator, the derivative of GP is still a GP. See details in (https://herbsusmann.com/2020/07/06/gaussian-process-derivatives/) and GPML textbook Chapter 9. For constructing GP given the observation, see details in appendix B. Basically, we assume $(s_t, \dot{s}_t)$ follows a Gaussian distribution (by GP), and hence conditioning on observation, we can get another Gaussian distribution.
>
>
> 8. > It is not clear what are related observations in general.
>
> Related (or correlated) observations refer to observations may be correlated. This is related to the notion of random effects in statistical modeling. Observations that have a underlying grouping structure will display such correlation. For instance, the HWD+ dataset extends the MNIST dataset by incorporating subject IDs and covariates. In our implementation, we sample training data within each subject, enabling GP-CFM to preserve subject-level correlations (grouping effects) for "0-8-6" across three observations. Another example is time-series data, as demonstrated in the new LFP application. For details on the LFP application, refer to General Comments 3 and Appendix G in the updated manuscript.
>
> 9. > Such a formulation seems too artificial (and beyond the scope of the classical FM problem formulation).
>
> We believe that incorporating the classical notion of random effects into the context of generative models is an important step to allow such algorithms like FM to incorporate rich experiment designs in practical applications. In many experiments, observations are related by grouping structure. For example, the HWD+ dataset includes subject IDs, and there is a subject-level correlation (grouping effect) across three digits. While I-CFM cannot capture the grouping effect across three observations, GP-CFM can. A more illustrative example is time-series data, demonstrated through the newly added LFP example. For details, see General Comments 3 and Appendix G in the updated manuscript.
>
> 10. > What is the relationship between your method and OT-CFM?
>
> Our stream-level model is more general, as GP-CFM and OT-CFM have different modeling goals and are complementary.
>
> Specifically, as discussed in Section 3.1, the stream model can be separated into two parts: 1) marginal model on endpoints $\pi(x_0, x_1)$ and 2) conditional model of $s$ given endpoints $p_{s}(\cdot\mid x_0, x_1)$. The OT-CFM focus on $\pi(x_0, x_1)$, and jointly samples pairs of endpoints using the 2-Wasserstein optimal transport (OT) map, while GP-CFM focuses on $p_{s}(\cdot \mid x_0, x_1)$, designing it with a GP. These two strategies can be combined, as demonstrated in the MNIST data application (GP-OT-CFM, see Appendix F).
>
> 11. > Can you explain in a little more detail how you found the FID for MNIST?
>
> For the MNIST dataset, FID is calculated using custom code. The 299x299 resolution is achieved by upsampling MNIST images, which is necessary for FID calculation because the Inception model used for FID requires 3-channel RGB images at a resolution of 299x299. This upsampling converts the original grayscale 28x28 MNIST images into a compatible format, ensuring accurate feature extraction and reliable FID scores. For more details on FID calculation, the clean-fid library (https://github.com/GaParmar/clean-fid?tab=readme-ov-file), which we used for the CIFAR-10 calculations, serves as a useful reference.

---

> ### Comment · Reviewer_yJK9 · 2024-11-30
>
> I thank the authors for their detailed response.
>
> I have carefully read the updated version of the paper, the responses to other reviewers.
>
> At this point, I would like to emphasise that the paper addresses two fundamental problems.
> The first -- the problem that classical generative models solve. Namely, to be able to sample more samples from an unknown distribution. This problem is solved in the CFM approach using various conditional maps. If the simplest I-CFM uses a linear map, then there are other approaches besides it, and nothing prevents from using any other map. It is also possible to take a stochastic map, e.g., Brownian Bridge. In fact, this is what the authors did by taking a stochastic map. But this approach is not new. A similar approach is considered, for example, in [1],[2], and in [3] the theoretical properties of vector field and score in the case of stochastic map are studied. Thus, the scientific novelty of the results presented in the paper is limited.
> As for experiments, as I wrote in the main review, only a limited number of cases are considered. The updated version of the paper includes Fig. 5C comparing FID on CIFAR-10 of this method and I-CFM. This graph cannot be considered convincing for several reasons:  I-CFM is the simplest method (there is no comparison with OT-CFM, Schrödinger bridge approaches, etc.), the numerical data are given without errors, and the numerical data themselves differ only slightly in the plots. But most importantly, in the paper on I-CFM ([4], see Fig. 5) for 1000 NFE the  FID is about 3--4, while in the graph shown the minimum FID is greater than 16.
>
> As for the second task -- it's essentially a time-series prediction. For this problem there are also approaches based, among others, on Schrödinger bridge and other approaches. The weak point of this paper is incorrectly chosen examples. Instead of taking data where the time parameter is natural from the point of view of the problem formulation (for example, the cell dynamics problem, which I mentioned in the main review), the authors considered too artificial problems, very far from real-world problems.
>
> Besides these main points, there are still minor ones: after the correction, the mathematical rigour is insufficient, the description of Algorithm 1 is not completed (it is still not clear how the sampling process occurs), I consider the way of calculating FID described in the replies to me as incorrect, etc. I recommend adding more rigour to the maths and running larger scale experiments on high dimensional data along with real-world data for time-series.
>
> Thus, I keep my score 3.
>
>
>
> [1] Gefei Wang, Yuling Jiao, Qian Xu, Yang Wang, Can Yang Deep Generative Learning via Schrödinger Bridge // Proceedings of the 38th International Conference on Machine Learning, PMLR 139:10794-10804, 2021.
> [2] Alexander Y. Tong, Nikolay Malkin, Kilian Fatras, Lazar Atanackovic, Yanlei Zhang, Guillaume Huguet, Guy Wolf, Yoshua Bengio. Simulation-Free Schrödinger Bridges via Score and Flow Matching // Proceedings of The 27th International Conference on Artificial Intelligence and Statistics, PMLR 238:1279-1287, 2024.
> [3] Gleb Ryzhakov, Svetlana Pavlova, Egor Sevriugov, Ivan Oseledets   Explicit Flow Matching: On The Theory of Flow Matching Algorithms with Applications // ICOMP-2024
> [4] Alexander Tong, Kilian FATRAS, Nikolay Malkin, Guillaume Huguet, Yanlei Zhang, Jarrid Rector-Brooks, Guy Wolf, Yoshua Bengio  Improving and generalizing flow-based generative models with minibatch optimal transport https://openreview.net/forum?id=CD9Snc73AW

---

### Author Response · Authors · 2024-11-22
**General Comments**

We thank all the reviewers for their constructive comments. Following their suggestions, we have updated our manuscript and included two additional experiments to demonstrate applications to high-dimensional image generation (CIFAR-10, Section 5.1) and time-series data (LFP, Appendix G). The Python implementation code has also been updated accordingly in the supplementary material. Detailed updates are provided in the revised manuscript, and below, we summarize the major changes.

## 1. Notations

We have revised the notations throughout the manuscript to make the writing more concise and easier to follow. The major changes are as follows:

- The subscripts of the expectation now represent random variables, rather than a mixture of random variables and distributions as in the previous version.
- We now define the stream as $s = {s_t : 0 \leq t \leq 1}$, where $s_t$ is a random variable denoting the location of the stream $s$ at time $t$. Using $s_t$ instead of $x_t(s)$ makes the writing more straightforward and easier to understand.

## 2. CIFAR-10

Details are provided in Section 5.1. We compare the performance of I-CFM and GP-I-CFM in generating images from standard Gaussian noise. Visually, images generated by GP-I-CFM are generally sharper and exhibit more details compared to those from I-CFM (e.g., first row of Figure 5A). This observation aligns with the FID reported in Figure 5C, albeit at a slightly higher computational cost per iteration (I-CFM: ~3.6 iterations/s; GP-I-CFM: ~3.0 iterations/s) and requiring more iterations to converge. To potentially achieve a more significant performance improvement with GP-I-CFM, a shorter SE bandwidth can be used. This adjustment allows the algorithm to explore wider regions of the sampling space during CFM training (as illustrated in Figure 1A) and reduces sampling variance. However, it also leads to slower convergence. Practitioners should select the GP tuning parameters to balance computational time (number of iterations) and sample quality.

## 3. LFP

The details are provided in Appendix G. To demonstrate how GP-CFM can be applied to time-series data, we use covariate dependent GP-I-CFM on a session of local field potential (LFP) data from seven regions of the mouse brain. The session contains 214 repeated trials. To visually highlight temporal smoothness over time, we subset the data to include five time points. In summary, the training data consists of 214 observations, with each observation having dimensions of $5 \times 7$.

---

### Meta-Review · Area_Chair_XAq8 · 2024-12-23

**Metareview:**

This paper proposes a method for enhancing Conditional flow matching (CFM) by taking a Bayesian decision theoretic perspective. In specific, the authors suggest using Gaussian Processes (GPs) to model streams, i.e. instances of latent stochastic paths.

__Strengths__
* The paper is generally well-written
* Using the properties of GPs to model streams brings computational and analytical advantages, and opens up the way for other types of interesting modeling.

__Weaknesses__
* Significance: If we take this to be a predominantly theoretical paper, according to the reviewers it does not seem to be rigorous enough. On the other hand, if we take this to be a predominantly practical framework, there is the concern that the experimental results are not convincing enough.
* Novelty. The novelty of this paper is limited. One of the reviewers points out that the key ingredient of this proposal is introducing a stochastic map, an idea that has been explored extensively in the past, whereas another reviewer argues about similar methods to define interpolants.

Overall, the paper is not ready for publication, although I highlight the helpful suggestion by one of the reviewers: for a future version of this paper it might be a good idea to focus on the usage of GPs and double down on the flexibility aspects it can provide.

**Additional Comments On Reviewer Discussion:**

There has been very active discussion with the reviewers and authors being very engaged. Several clarifications and notational issues were discussed and to a large extent addressed. Several points regarding the novelty and claims of the paper were discussed, and the reviewers evidently read the rebuttal and updated version - however they remained unconvinced and explained in detail their reasoning.

---

### Decision · Program_Chairs · 2025-01-22

Reject